# Comparison of lightning activity in the two most active areas of the Congo Basin

Jean K. Kigotsi[1, 2], Serge Soula[1], Jean-François Georgis[1]

[1]Laboratoire d'Aérologie, Université de Toulouse, CNRS, Toulouse, France

[2]Département de Physique, Faculté des Sciences, Université de Kinshasa, République Démocratique du Congo

*Correspondence to*: Jean K. Kigotsi (jeankigotsi@gmail.com); Serge Soula (serge.soula@aero.obs-mip.fr)

**Abstract.** A comparison of the lightning activity in the two most active areas (Area_max for the main maximum and Area_sec for the secondary maximum) of the Congo basin is made with data obtained by the World Wide Lightning Location Network (WWLLN) during 2012 and 2013. Both areas of same size ($5° \times 5°$) exhibit flash counts in a ratio of about 1.32 for both years and very different distributions of the flash rate density (FRD) with maximums in a ratio of 1.94 and 2.59 for 2012 and 2013, respectively. The FRD is much more widely distributed in Area_sec, which means the whole area contributes more or less equal to the lightning activity. The diurnal cycle is much more pronounced in Area_max than in Area_sec with a ratio between the maximum and the minimum of 15.4 and 4.7, respectively. However, the minimum and maximum of the hourly flash rates are observed roughly at the same time in both areas, between 07:00 and 09:00 UTC and between 16:00 and 17:00 UTC, respectively. In Area_sec the proportion of days with low lightning rate (0-1,000 flashes per day) is much larger (~45% in 2013) compared to Area_max (~23% in 2013). In Area_max the proportion of days with moderate lightning rate (1,001-6,000 flashes per day) is larger (~68.5% in 2013) compared to Area_sec (~46% in 2013). The very intense convective events are slightly more numerous in Area_sec. In summary, the thunderstorm activity in Area_sec is more variable at different scales of time (annually and daily), in intensity and in location. Area_max combines two favourable effects for thunderstorm development, the convergence associated with the African easterly jet of the Southern Hemisphere (AEJ-S) and a geographic effect due to the orography and the presence of a lake. The location of the strong convection in Area_sec is modulated by the distance of westward propagation/regeneration of MCSs in relation with the phase of Kelvin waves.

## 1 Introduction

According to several studies about the lightning climatology around the Earth, the Congo basin is considered as the most active region with either a large maximum, or two distinct ones (Christian et al., 2003; Williams and Sátori (2004), Albrecht et al., 2011, 2016, Cecil et al., 2014, Soula et al., 2016). Actually, the features of the maximum area depend on the spatial resolution considered in the calculation of the flash rate density (FRD) and the scale resolution in the graphic representation. Albrecht et al. (2016) performed a very detailed analysis of FRD thanks to Lightning Imaging Sensor (LIS) data around the Earth, by using several spatial resolutions. They showed the features of the maxima FRD strongly depend on the spatial resolution and on the duration of the period considered for the study. Thus, the location and the value of the first- and second-ranked maxima FRD stabilize when the period is longer. With the better resolution (0.1°) used in Albrecht et al. (2016), the second-ranked hotspot is always located around [28°E; 2°S] from five years of data. Furthermore, they showed most of the first ten lightning hotspots over the entire African continent, including the strongest ones, are located in Democratic Republic of Congo (DRC). By considering the maps of FRD in Albrecht et al. (2016), the existence of two regions of maximum activity in DRC is displayed but the non linear scale does not allow a quantitative comparison of the maximum values.

Cecil et al. (2014) provided two maps of lightning flash density from the Lightning Imaging Sensor (LIS) and Optical Transient Detector (OTD) data with different resolution, 0.5° and 2.5° and a non linear scale. With a 0.5° resolution, two maxima are distinguished in the region of Congo Basin and only one with a 2.5° resolution. Two separated maxima are also visible in the study by Christian et al. (2003) with a resolution of 0.5° and a non-linear scale of density. However, in the latter study, neither maximum remain throughout the year in considering the lightning activity with 3-month seasons. Recently, Soula et al. (2016) showed a very pronounced maximum in the annual and seasonal lightning flash density in the eastern Democratic Republic of Congo (DRC) from World Wide Lightning Location Network (WWLLN) data with a 0.1° resolution and a linear scale. In this study, a secondary maximum was also highlighted west of the main maximum, especially during the first part of the 9-year period of study. This secondary maximum was less pronounced and scattered ~~in~~ over a large area. In this study the region of maximum activity could be analyzed in detail because the linear scale for flash density was better adapted for large values compared to previous studies.

The results of Soula et al. (2016) provided the following characteristics. The main maximum in lightning flash density is observed every year in one small region of the DRC, at about 28°E and between 1°S and 2°S. This maximum is embedded within a region of large values of lightning flash density strongly contrasting with the whole study area. The geographical extent of this region is approximately 300 km north-south and 200 km east-west. It is located in the area where many authors identified the maximum of the planetary lightning activity, as Christian et al. (2003) who falsely attributed it to Rwanda, Cecil et al. (2014) and Albrecht et al. (2011). The high spatial resolution and the linear scale used in Soula et al. (2016) allowed a better localization and specification of its shape and amplitude characteristics. In addition, the maximum number of days with thunderstorms has been found in the same area (189 days of storms in 2013) as the average number of flashes per day of storms (approximately 8 flashes per day). Another area of large flash density considered as a secondary maximum was pointed out in Soula et al. (2016). This area was broader but less contrasting from year to year during the period of the study. It extends roughly from the centre of DRC to Congo to the west and to Angola to the south.

The goal of this study is to compare the characteristics of lightning activity in the two areas of maximum activity. The second section describes the data and the methodology used, the third section presents the results from several comparisons, and the fourth section is devoted to a discussion.

**2 Data and methodology**

By following the study by Soula et al. (2016), we define two areas of equal area, one for the main maximum considered as "Area_max" and the other for the secondary maximum considered as "Area_sec". They are identified by latitude and longitude values in the following intervals:

[25°E; 30°E] and [4°S; 1°N] for Area_max

[18°E; 23°E] and [4°S; 1°N] for Area_sec

We use data from the WWLLN for the present study. The WWLLN (www.wwlln.net/) is a global lightning detection network around the Earth. The electromagnetic radiation emitted by lightning strokes (from cloud-to-ground and intracloud flashes) at very low frequency (VLF) and called sferics are detected by the sensors of the WWLLN (Abarca et al., 2011). These strokes are then localized by using the time of group arrival technique (TOGA) (Dowden et al., 2002). The stations can be separated by thousands of km because VLF

frequencies can propagate within the Earth-Ionosphere wave guide with very little
attenuation. Since its implantation in March 2003, the WWLLN has been improved in terms
of number of stations and development of the processing algorithm (Rodger et al., 2008). In
order to give an idea of the growth of the number of WWLLN stations spread on the planet, it
was 11 in 2003, then 23 in 2005, 30 in 2007 and 67 in 2013, according to the report made by
Rodger et al. (2014). Indeed, the changes in the network during this 9-year period (2003-
2013) can explain the continuous increase of the detection efficiency (DE) observed by Soula
et al. (2016) in the total domain of the study. According to Abarca et al. (2011), DE for CG
flashes is about twice that for IC flashes.
We analyze the DE evolution during this period for each area. For this purpose and in the
same way as Soula et al. (2016) for the whole Congo basin area, Figure 1 displays the annual
numbers of lightning flashes detected by WWLLN and LIS in Area-max and Area_sec during
the period 2005-2013. In the same graph, the values of the WWLLN DE relative to the LIS
data, are reported for each area. DE is calculated by following the methodology developed by
Soula et al. (2016), i.e. by applying the correction coefficient for the estimation of the number
of the whole lightning flashes LIS could detect with a continuous survey. First, the number of
flashes detected by LIS in each area does not vary much during the period, it is always larger
in Area_max, its minimum is observed for 2007 in each area and more pronounced for
Area_sec, and the maximum is observed for 2005 in each area too. Thus, no increase
tendency is observed in each area. Secondly, the number of flashes detected by WWLLN in
each area increases after 2008, especially during the last two years 2012 and 2013. As a
consequence, DE is significantly larger for 2012 and 2013, and reaches 4.96% and 7.50% in
Area_max, respectively, and 4.24% and 6.11% in Area_sec. This increase of DE is
completely independent of the number of flashes detected by LIS that is relatively stable
during the last years, which means it is totally related to the WWLLN performance.
According to the DE values, we select the last two years of the period (2012 and 2013) for a
comparison of the characteristics of the lightning activity in Area_max and Area_sec.
**3 Results**
**3.1 Spatial distribution of the lightning activity**
Figure 2a-b shows the annual FRD, in flash $km^{-2}$ $yr^{-1}$, calculated with a resolution of 0.05°
from WWLLN data in the large domain of the Congo basin for 2012 and 2013, respectively.
Figure 2c-d shows the number of days of the year with lightning activity in the same domain
with the same resolution for 2012 and 2013, respectively. The white frames indicate the two
areas with strong activity (left Area_sec and right Area_max). Table 1 displays the flash
count, the maximum FRD for both areas and for each year. Both areas of same size ($5° × 5°$)
exhibit total flash counts in a ratio of about 1.32 for both years, which indicates a stable
situation from one year to the next. On the contrary, the ratio of the maximum FRD is very
different from one year to the next, since it is 1.94 and 2.59 for 2012 and 2013, respectively.
This difference can be easily understood since the maximum value is very localized and can
change substantially from one year to the next, and furthermore the spatial density resolution
used in the study is very high, with a value of $0.05°$. The maximum value of the density
depends on the spatial resolution, in the sense that it increases when the resolution becomes
higher. By comparing with the values reported by Soula et al. (2016) at a resolution of $0.1°$, it
is clear that the maximum of the annual FRD is larger for $0.05°$. Indeed, it is 12.86 fl km$^{-2}$ yr$^{-1}$
at $0.1°$ and 15.33 fl km$^{-2}$ yr$^{-1}$ at $0.05°$ in 2013, and it is 8.22 fl km$^{-2}$ yr$^{-1}$ at $0.1°$ and 8.62 fl km$^{-2}$
yr$^{-1}$ at $0.05°$ in 2012. On the other hand, the maximum number of stormy days is lower with
the resolution of $0.05°$, from 189 to 125 days for 2013 and from 167 to 99 days for 2012. This
observation is consistent since a day is stormy when at least one flash is detected in the pixel.
The difference between the distributions in the two areas clearly appears regarding both
lightning FRD and number of days of the year with lightning activity in Figure 2. Indeed, the
highest values of both parameters are located in the same region of the $5° × 5°$ frame for
Area_max while they are much more scattered in the frame for Area_sec. The difference
between both areas is stronger for FRD compared to the number of days with thunderstorms,
which means that the number of flashes per stormy day is larger for Area_max. It means that
the storms in Area_max are more active and/or more stationary, and/or more numerous (Soula
et al., 2016). The differences observed in the maximum values and the distributions of the
lightning FRD indicate specific conditions for the thunderstorm development in Area_max.
These conditions are the presence of a mountain range that exceeds 3000 meters ($28.75°E$;
$1.5\text{-}2.2°S$), on the west side of which the FRD increases markedly, and the presence of the
lake Kivu ($29.2°E$; $1.9°S$) above which the FRD increases (Soula et al., 2016). No specific
shape of the FRD or stormy day is visible in Area_sec.
**3.2 Daily cycle**
Figure 3 shows the daily cycle of the flashes detected by the WWLLN in Area_max and
Area_sec, for 2012 and 2013. The time is indicated in UTC, which is two hours late compared
to Local Time (LT = UTC + 2). These flash counts are calculated over one hour and averaged
over all days of the year. The time scale of the graph is made so that the flashes are associated
with the beginning of the 1-hour period of calculation. Both areas exhibit the same type of
diurnal lightning activity with a large proportion of flashes during the afternoon and whatever
the year. The minimum and maximum numbers of flashes are observed roughly at the same
time in both areas. The minimum is observed in the morning between 08:00 and 09:00 UTC
for Area_max and between 07:00 and 08:00 UTC for Area_sec, for both years. The maximum
is observed in the afternoon, between 16:00 and 17:00 UTC for Area_max and for both years
and for Area_sec in 2013, and between 17:00 and 19:00 UTC for Area_sec in 2012. The
contrast in flash counts between the morning minimum and the afternoon maximum is
stronger for Area_max (ratio of 14.5 and 15.4, for 2012 and 2013, respectively) than for
Area_sec (ratio of 6.2 and 4.7, for 2012 and 2013, respectively). It means the diurnal cycle is
much more pronounced in Area_max. Consequently, while the lightning flash rate is larger in
Area_max for the main part of the day, it is lower during a short interval between 06:00 and
10:00 UTC corresponding to the minimum activity in both areas.
**3.3 Day-to-day variability**
We compare the lightning activity in both areas in terms of daily distribution of flashes
detected during one year. The years of reference are 2012 and 2013 with a total of 366 and
362 days, respectively, available in the database. The flash count is performed day by day in
each area and then the days are classified by range of flash numbers. Thus, Table 2 displays
the result of the classification for each area and each year, over 12 classes of flash number.
This result is expressed in terms of number of days for each area and year, and in proportion
(%) of the total number of days for the year in each area. The incrementing of each class is
done on 1,000 flashes, except for the class CL1 that is on 900 flashes from 101 to 1,000
flashes. The first class CL0 corresponds to 0-100 flashes to distinguish the days with a very
low number of flashes. The last class CL11 groups the days with more than 10,000 flashes. To
make easier the interpretation of the results, they are also plotted in Figure 3.
The distribution is similar for both years, (a) for 2012 and (b) for 2013. The number of days in
CL0 is much larger for Area_sec than for Area_max (59 and 7, respectively, in 2012, 43 and 4
in 2013), as indicated in Table 2. For CL1 corresponding to the flash numbers 101-1,000, the
number of days is also larger for Area_sec, slightly in 2012 with 130 and 121 days,
respectively, markedly in 2013 with 121 and 80 days, respectively. On the contrary, the
number of days for classes corresponding to intermediate flash numbers (CL2 to CL4 in 2012,
CL2 to CL6 in 2013) is significantly larger for Area_max, for both the cumulative number of
days (202 against 144 in 2012 and 248 against 168 in 2013) and for each class considered
separately. For the classes with a very high activity (CL5 to CL11 and CL7 to CL11, in 2012
and 2013, respectively) the total number of days is small and not very different in both areas
(36 and 30 in 2012, 20 and 30 in 2013, for Area_max and Area_sec, respectively).
From 2012 to 2013, for both areas, the proportion of the number of day decreases in the first
three classes (CL0-CL2) and for the cumulative value it is ~62% in 2012 and ~45% in 2013
for Area_max, and ~70% in 2012 and ~61% in 2013 for Area_sec. It is almost equal in CL3:
~20% in 2012 and ~19% in 2013 for Area_max, and ~14% in 2012 and ~14% in 2013 for
Area_sec. It increases almost in all classes after CL3 and for cumulative value it is ~18% in
2012 and ~36% in 2013 for Area_max, and ~16% in 2012 and ~25% in 2013 for Area_sec.
**3.5 Correlation between daily lightning activities**
Now we consider the lightning activity for a comparison day by day of both areas to perform
a quantitative correlation. The goal is to evaluate if both areas are simultaneously concerned
by the storm activity or if they are with a shifted time. In order to illustrate the result about
this correlation between lightning activity in Area_max and Area_sec, we display the graph of
correlation between the daily lightning flash amounts for both areas and in 2013. These daily
counts are calculated in two ways, first by considering the calendar day (00h00 – 24h00 UT)
and then according the daily cycle of lightning activity between two minimums (06h00 –
06h00 UT, see Figure 2). Figure 5 shows the result of this correlation study: (a) for the
calendar days and (b) for the lightning cycle days.
In the first case the correlation coefficient $R^2$ is ~0.118 and in the second case it is ~0.064.
Thus, the correlation is weak but positive, that is to say the tendency is that when the daily
flash number increases for one area it also increases for the other. At first glance, both
distributions are similar. They reflect the trend highlighted by Figure 4 insofar as the low
values (≤ 1000 flashes per day) are more numerous in Area_sec. Inversely, the intermediate
values (between 1,001 and 5,000 flashes per day) are more numerous in Area_max with 230
days in 2013, against 156 days for Area_sec. For the values exceeding 10,000 flashes per day,
there are 7 days for Area_max and 5 days for Area_sec in 2013 (Figure 5a). In Figure 5b,
these values are 6 and 8, respectively, which means there are more days with a large number
of lightning flashes in Area_sec, by considering the daily cycle of the lightning activity. This
observation is consistent with the fact that the lightning activity is more widely distributed
during the day in Area_sec as indicated in Figure 3. This may be due to the contribution of
nocturnal lightning by mesoscale convective systems (MCSs) or isolated storms that develop
later in the afternoon if compared to Area_max. Indeed, the work by Albrecht et al. (2016)
shows in their Figure 3 that during the night, the hotspots located in Area_sec (i.e, 6th and 7th
Africa's hotspots) exhibit a larger contribution to the daily lightning activity. Thus, by
considering the day according the lightning activity (06h00-06h00), the episodes of strong
lightning activity in this area are more likely to be counted in full.
**3.6 Month-to-month variability**
Figure 6a-b shows the monthly proportion of flashes detected in Area_max and Area_sec
during 2012 and 2013. The minimum proportion is found in August and in Area_sec (between
3 % and 4 %) for both years. The maximum proportion is also found in Area_sec in May for
2012 (about 14%) and in December (about 14%) for 2013. These two characteristics show
that the variability is always stronger in Area_sec than in Area_max although the distribution
is different from 2012 to 2013 for both areas. For example, in April it is 6.1% and 11.3% for
Area_max, 5.7% and 9.4% for Area_sec, in 2012 and 2013, respectively. Inversely in May,
the proportion of each area is much lower in 2013 compared to 2012 (4.7% and 8.1% for
Area_max, 7.9% and 13.9% for Area_sec). For a given month, the respective proportions for
Area_max and Area_sec remain in the same order, except for the first three months of the
year.

242       Figure 6c shows the 3-month proportion over a longer period including data from
2011. The 3-month periods are chosen according to Christian et al. (2003), Jackson et al.
(2009), and Soula et al. (2016). Thus, the months of June, July and August are grouped in
JJA, September, October and November in SON, December, January and February in DJF,
and March, April and May in MAM. The annual variability at this 3-month scale is more
visible and constant from one year to the next. Indeed, for both areas, the minimum is always
in JJA with a constant decrease during the preceding 3-month periods. For the maximum, it
seems SON is more favourable to Area_max while DJF is for Area_sec.
**4 Discussion**
Albrecht et al. (2016) studied the lightning hotspots over the Earth, based on satellite optical
observations of lightning. They consider that a hotspot is a region 100-km in radius around a
maximum of FRD. They found that six out of the ten most active spots over the whole
African continent, including the three strongest ones, are located in an area corresponding to
Area_max while only two are located in an area corresponding to Area_sec. Our results
confirm the predominance of the larger FRD in Area_max.
The characteristics of the diurnal cycle observed in Area_max and Area_sec is consistent
with Laing et al. (2011). These authors analyzed the cycle of the deep convection over a large
area of tropical Africa including both areas of our study and during 2000-2003. For two 1-
hour intervals (14:00-15:00 UTC and 17:00-18:00 UTC) besides eight considered in their
study, they found the location of a sharp maximum of the average hourly frequency of coldest
clouds in eastern DRC close to Area_max. The intervals 15:00-16:00 and 16:00-17:00 UTC
were not plotted in their graphs. They noted this maximum for the two months April and
October analyzed in the study. They also showed that the thunderstorm activity is minimum in
the part of DRC that corresponds to both areas of our study during the time interval 05:00-
06:00 UTC in April and during 08:00-09:00 UTC in October (06:00 and 07:00 UTC were not
plotted). The present observations about minimum and maximum lightning activities
displayed in Figure 2 are consistent with those by Laing et al. (2011). Indeed, the maximum
of the activity is invariably between 16:00 and 17:00 UTC for Area_max, and in a larger
temporal window for Area_sec (~17:00-19:00 UTC in 2012 and 16:00-17:00 UTC in 2013).
The maximum storm activity is therefore more variable in time for Area_sec. The minimum is
invariably between 07:00 and 08:00 UTC for Area_sec, between 08:00 and 09:00 UTC for
Area_max. In Albrecht et al. (2016) for the study of lightning hotspots, the daily cycles are
considered for several hotspots located in our areas. They found a daily cycle more pronounced
for the hotspots included in Area_max compared to the hotspots included in Area_sec, which
is consistent with the present study.
The comparison of the monthly activity in Area_max and Area_sec in 2012 and 2013
suggests that the seasonal contrast is stronger in Area_sec where the maximum monthly
amounts are observed in May and December respectively, and the minimum in August for the
two years. At the seasonal scale, the monthly activity is cumulated over three months
following the average monthly activity found in Soula et al. (2016) for the whole Congo
basin. The inter-annual variability is well visible and reproduced from one year to the next.
Even in these three years the minimum proportion is always in August and in Area_sec (about
3 to 4%). The maximum proportion is also in Area_sec but on different months (from 14 to
16%). So the seasonal contrast is much stronger in Area_sec than in Area_max. This result,
due to the migration of the Intertropical Convergence Zone (ITCZ), is consistent with the
contrast of the seasonal variation in lightning activity found in Soula et al. (2016). Area_max
is less impacted by the migration of the ITCZ because the triggering of thunderstorms in this
area has a very local origin.
The positive correlation observed between the daily activities of the two areas means there
may be an influence between them or a common cause to explain the storm activity. However,
the low value of the correlation coefficient indicates the activities can be different on the
quantitative aspect. Figure 7 displays the daily density of lightning flashes detected by
WWLLN on 25[th] of December 2013 in Area_sec (a) and in Area_max (b). This day is
considered because the activity is strong in both areas with 18107 and 10257 flashes detected
in Area_sec and Area_max, respectively. Firstly, this distribution shows the lightning density
is high (scale in fl $km^{-2}$ $day^{-1}$) in local spots that correspond to convective cores of
thunderstorms. In other words, for a given day, the lightning activity can be strong in a
restricted area and weak around in term of flash number. This characteristic of the storm
activity is well known and pointed out by many works (Carey et al., 2005; Soula et al., 2014).
Secondly, the lightning spots seem east-west elongated in majority, which could indicate a
propagation of the storms within this direction. Thus, the strong activity of a given storm is
probably limited over the time. However, the correlation between both areas probably exists
because of the eastward propagation of conditions favourable to the development of
thunderstorms, as instability of the atmosphere. Indeed, Laing et al. (2011) showed
convection over equatorial Africa can be modulated by different conditions at synoptic scale
for local occurrence or propagation of mesoscale convective systems. They especially
mentioned the eastward-moving equatorially trapped Kelvin waves, the south-westerly
monsoonal flow and the midlevel easterly jets. It is therefore consistent to obtain a low
correlation between our two areas characterized by a strong annual storm activity.
Furthermore, the correlation study is done at the scale of the day and as most thunderstorms
develop at the end of the day, storm activity can occur during the following day in Area_sec
that is several hundred kilometres to the West.
The distribution of storms in the Congo Basin mainly results from four contributions,
namely: development, propagation, merging and regeneration of thunderstorms. As
thunderstorms can develop everywhere in the Congo basin, they can naturally form in both
Area_max and Area_sec. However, the great lakes and numerous mountains of Rift valley
close to Area_max offer most favourable conditions for development and enhancement of

thunderstorms. The most intense storms, at planetary scale, are found in the Congo Basin (Zipser et al., 2006). Area_max is probably the most active region in the world in terms of thunderstorms since the number of days of the year with thunderstorm activity is found to be maximum there (Figure 1c-d) and the density of lightning is large over this extended area (Soula et al., 2016). On the other hand, according to previous studies, Equatorial Africa thunderstorms spread from the east to the western Congo basin (Laing et al., 2011; Nguyen and Duvel, 2008; Laing and Fritsch, 1993). Then thunderstorms may propagate from Area_max to Area_sec but different processes as merging and regeneration may affect their intensity and induce different characteristics in these areas. Several studies have shown that heterogeneity of soil moisture or vegetation play a role in thunderstorms triggering (Taylor et al., 2011; Garcia-Carreras et al., 2010). Furthermore, the modelling results of the Global Land Atmosphere Coupling Experiment (GLACE) classified Equatorial Africa, including Area_max and Area_sec, among the regions of strong coupling between the atmosphere and the soil moisture (Koster et al., 2004). Thus, differences of soil moisture and/or vegetable cover between Area_max and Area_sec may contribute to the differences between lightning activities of the two areas.

Farnsworth et al. (2011) pointed out that the MCSs constitute the fundamental unit of vertical energy transport in Central Africa. In other words, convection in this region generally leads to the formation of MCSs. This observation is consistent with the results of Liu and Zipser (2005) and Zipser et al. (2006) (on deep convection in the Congo basin). They showed convection in the Congo basin frequently overshoots the tropopause. The climatology of MCSs in Equatorial Africa, including the whole Congo basin, was presented in Jackson et al. (2009). From a five-year series of data, these authors have shown that the zone on horseback at the equator between 5°S and 5°N and extending from the Atlantic coast to the west side of the high mountains of the Rift Valley is the most active in terms of storm activity because it includes two of four maxima in the number of MCSs that they have identified. In our study, Area_max and Area_sec coincide with the region where Jackson et al. (2009) found the main number maximum of MCS. Actually, in Jackson et al., two cores appeared in the structure of this main maximum, one that corresponds to Area_sec with a less pronounced maximum of number of MCS and a larger number of lightning flashes per MCS. The second core in Jackson et al. corresponds to Area_max with a more pronounced maximum. They explain the origin of the large number of MCS in this large area by a maximum of midtropospheric convergence on the west side of the African easterly jet of the Southern Hemisphere (AEJ-S).

They observe this condition more pronounced in SON season compared to MAM in the same
way that we observe also more flashes according to Figure 6c. Indeed, according to Mohr and
Thorncroft (2006) and Laing et al. (2008), the vertical shear related to the African easterly jet
(AEJ) influences the location of intense convective systems. Furthermore, mountain ranges
help to initiate long-lived MCSs (Laing et al., 2008; 2011). According to these authors, in all
the regions the convection initiates over the elevated terrain and then propagates in conditions
of moderate vertical shear to develop into mesoscale systems. On the other hand and
according to several authors, the propagation of convection in Equatorial Africa is modulated
by convectively coupled, equatorial Kelvin waves (Laing et al., 2011). During the active
phase of these eastward-propagating large-scale waves, MCSs are larger and more intense.
These convection systems occur farther east from day to day, and propagate westward within
the Kelvin wave envelope. During the dry phase of the Kelvin waves an upper-level
convergence is produced, which eliminates the deep convection and the westward
propagation. Thus, the region corresponding to Area_max seems to have a stronger maximum
of MCS number, as we find a larger FRD. Area_max combines two conditions favourable for
thunderstorm activity, the convergence evoked by Jackson et al. (2009) for the large region
and a local orographic effect that reinforces the effect of the first one. Area_sec seems to take
advantage of the westward propagation/regeneration of MCS, at a distance from the initial
occurrence that depends on the phase of the Kelvin waves, which explains the widespread
large values of FRD observed within this area.
The presence of mountains or elevated terrain is always a determining factor in the
mechanism of thunderstorm. For example at a very local scale, Munoz et al. (2016) explain
the role of the topography combined with Nocturnal Low Level Jet in the largest FRD in the
world observed in the region of the lake Maracaibo, Venezuela. At a more global scale,
William and Sátori (2004) compared the lightning and rainfall activities in both Amazone and
Congo basins and interpret the greatest FRD observed in Congo basin in terms of features
more continental (drier and warmer) and a larger elevation.
According to Zipser et al. (2006) the proportion of intense convective events is larger in the
region corresponding to Area_sec compared to that corresponding to Area_max (see their
figure 3). This result is consistent with the present figure 5 concerning the distribution of the
daily flash number in each area, especially the graph (b) where the flash counts are made from
06:00 to 06:00 UTC. Furthermore, the DE is a little lower in Area_sec compared to
Area_max, according to the results displayed in Figure 1. Thus, Area_sec is concerned by a
more irregular thunderstorm activity, with both the least active days and the most active days.
It is well illustrated with the example in Figure 7, displaying the daily lightning activity for
the most active day in Area_sec (see Figure 5a). Indeed, the FRD for the day is more scattered
in the whole area for Area_sec. The distribution of thunderstorm activity is substantially
different in each area, concentrated with a very marked daily cycle in Area_max, and
scattered with a daily cycle much less pronounced.
**5 Conclusion**
The spatial and temporal characteristics of the lightning activity are analysed in two areas of
the Congo basin, Area_max with the strongest thunderstorm activity and Area_sec with a
secondary maximum. First, the lightning flashes are much more concentrated in the same part
of Area_max for both years, while they are widespread in Area_sec. Secondly, the frequency
of days with low activity is larger in Area_sec and the frequency of days with high activity is
larger in Area_max. However, the frequency of days with very high activity is similar in both
areas and even the largest daily flash numbers are detected in Area_sec. Thirdly, a stronger
contrast between the maximum and the minimum in the daily cycle is observed in Area_max
with a ratio of about 15.4 while it is only 4.7 for Area_sec. In conclusion, the thunderstorm
activity is more variable in Area_sec, in terms of location, daytime of occurrence, seasonal
distribution and intensity in terms of number of flashes. These differences are consistent
because Area_max combines two favourable effects for thunderstorm development, the
convergence associated with the AEJ-S, especially during SON and DJF, and a geographic
effect due to the orography and the presence of a lake. The location of the strong convection
in Area_sec is widespread, according to the distance and direction of
propagation/regeneration of MCSs that initiate farther eastern, especially in relation with the
phase of Kelvin waves.
**Acknowledgments**
The authors thank the World Wide Lightning Location Network (http://wwlln.net/) and
Christelle Barthe from University of la Réunion (France) for providing the lightning location
data used in this study. JK is grateful to the French "Ministère des Affaires Etrangères", to the
French Embassy in DRC, especially Patrick Demougin, for supporting his stay in France and
"Groupe International de Recherche en Géophysique Europe/Afrique" (GIRGEA) for
cooperation organisation, especially Christine Amory. JK thanks Professor Albert Kazadi
from University of Kinshasa for his support, help and discussions.

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

**Table 1.** Flash count and flash density in both areas.

| | Flash count | | Maximum flash density $(fl\ yr^{-1}\ km^{-2})$ | |
|---|---|---|---|---|
| | 2012 | 2013 | 2012 | 2013 |
| Area_max | 696,144 | 1,000,687 | 8.6 | 15.3 |
| Area_sec | 526,278 | 760,405 | 4.4 | 5.9 |
| ratio | 1.32 | 1.32 | 1.94 | 2.59 |


**Table 2.** Number of days corresponding to lightning classes in the two study areas during the
2012 (366 days) and 2013 (362 days). The percentage is calculated in relation to the total
number of days during the year.

| Flash number | CLASS | Number of days (%) | | | |
|---|---|---|---|---|---|
| | | 2012 | | 2013 | |
| | | Area_max | Area_sec | Area_max | Area_sec |
| 0 – 100 | CL0 | 7 (*1.91*) | 59 (*16.12*) | 4 (*1.10*) | 43 (*11.88*) |
| 101 – 1,000 | CL1 | 121 (*33.06*) | 130 (*35.52*) | 80 (*22.10*) | 121 (33.43) |
| 1,001 – 2,000 | CL2 | 99 (*27.05*) | 68 (*18.58*) | 79 (*21.82*) | 58 (*16.02*) |
| 2,001 – 3,000 | CL3 | 73 (*19.94*) | 52 (*14.21*) | 70 (*19.34*) | 52 (*14.36*) |
| 3,001 – 4,000 | CL4 | 30 (*8.20*) | 24 (*6.56*) | 43 (*11.88*) | 29 (*8.01*) |
| 4,001 – 5,000 | CL5 | 16 (*4.37*) | 17 (*4.64*) | 38 (*10.50*) | 17 (*4.70*) |
| 5,001 – 6,000 | CL6 | 10 (*2.73*) | 7 (*1.91*) | 18 (*4.97*) | 12 (*3.31*) |
| 6,001 – 7,000 | CL7 | 4 (*1.09*) | 4 (*1.09*) | 12 (*3.31*) | 11 (*3.04*) |
| 7,001 – 8,000 | CL8 | 2 (*0.55*) | 1 (0.27) | 7 (*1.93*) | 10 (*2.76*) |
| 8,001 – 9,000 | CL9 | 4 (1.09) | 1 (0.27) | 2 (*0.55*) | 2 (*0.55*) |
| 9,001 – 10,000 | CL10 | 0 (*0.00*) | 0 (*0.00*) | 2 (*0.55*) | 2 (*0.55*) |
| > 10,000 | CL11 | 0 (*0.00*) | 0 (*0.00*) | 7 (*1.93*) | 5 (*1.38*) |
| Total | | 366 (*100*) | 366 (*100*) | 362 (*100*) | 362 (*100*) |









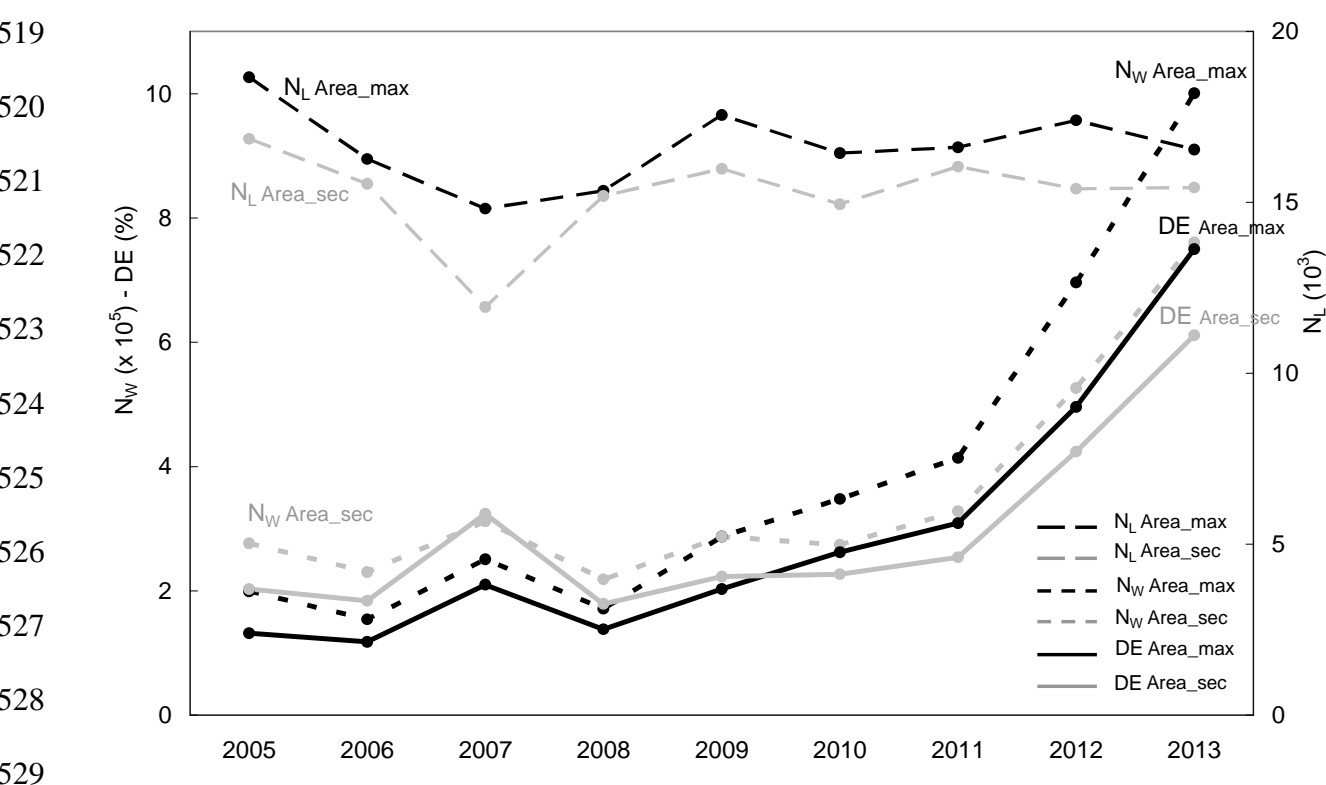

Figure 1. Annual number of flashes detected by the WWLLN ($N_W$) and that detected by LIS
($N_L$) for each area, and estimated detection efficiency (DE) for WWLLN data relative to LIS
data, according to the methodology developed in Soula et al. (2016).










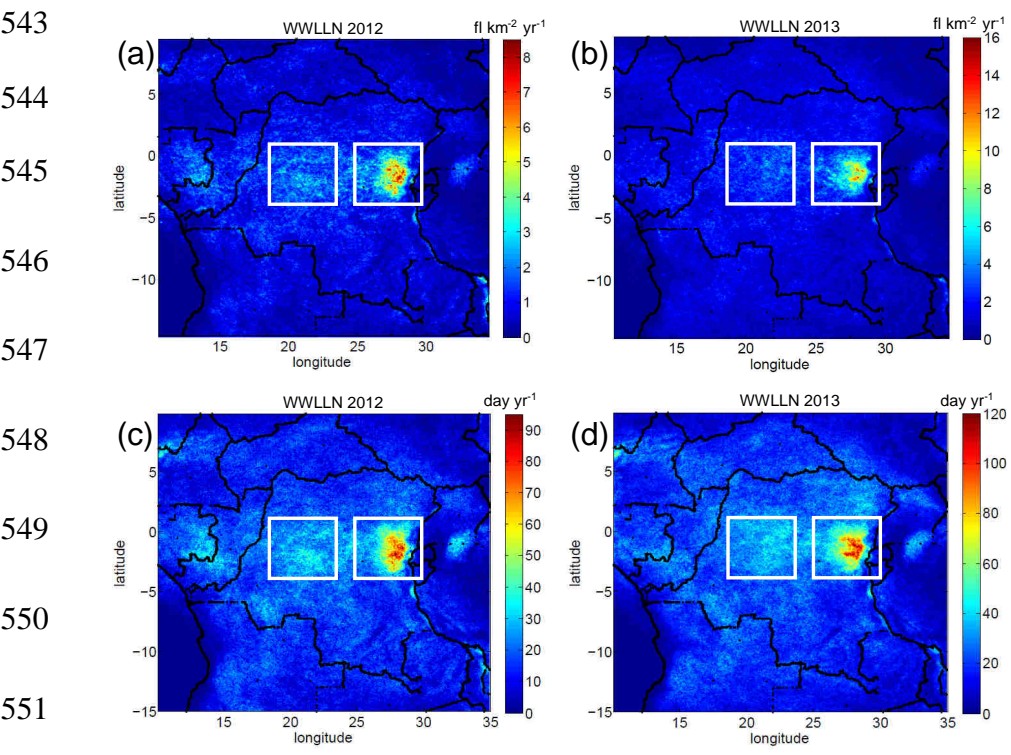

Figure 2. (a) and (b) Lightning density in fl km$^{-2}$ yr$^{-1}$ calculated at a resolution of 0.05° from
WWLLN data in the area of Congo Basin for 2012 and 2013, respectively. (c) and (d)
Number of days of the year with thunderstorm activity in the same area with a resolution of
0.05° for 2012 and 2013, respectively. The white frames indicate the two zones with strong
activity (left Area_sec and right Area_max).

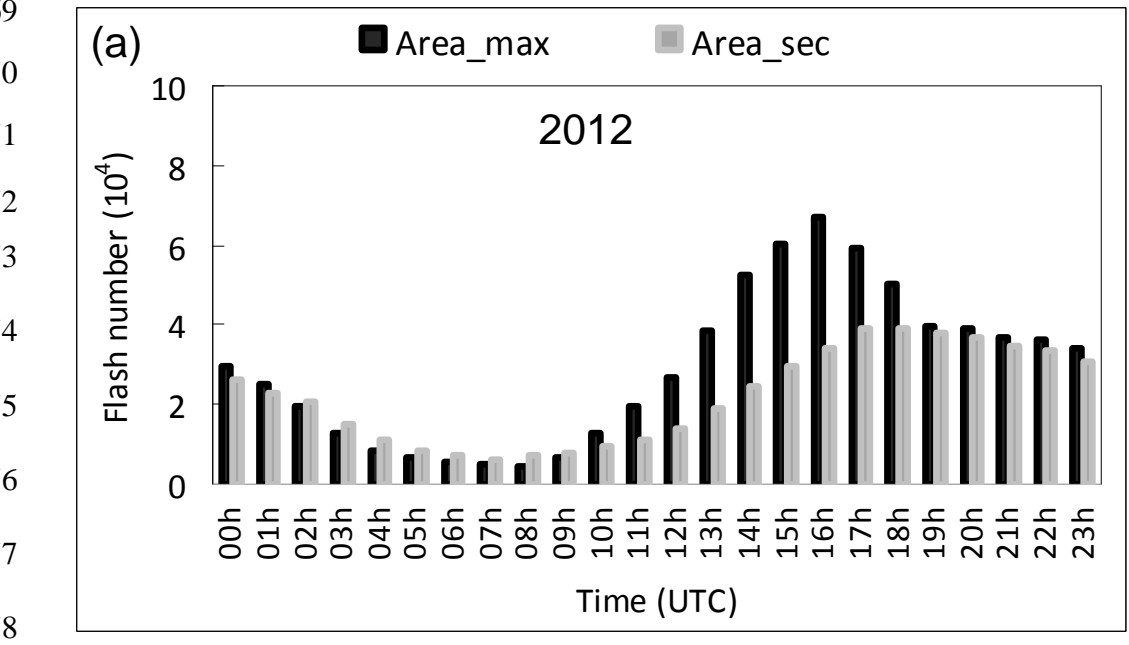

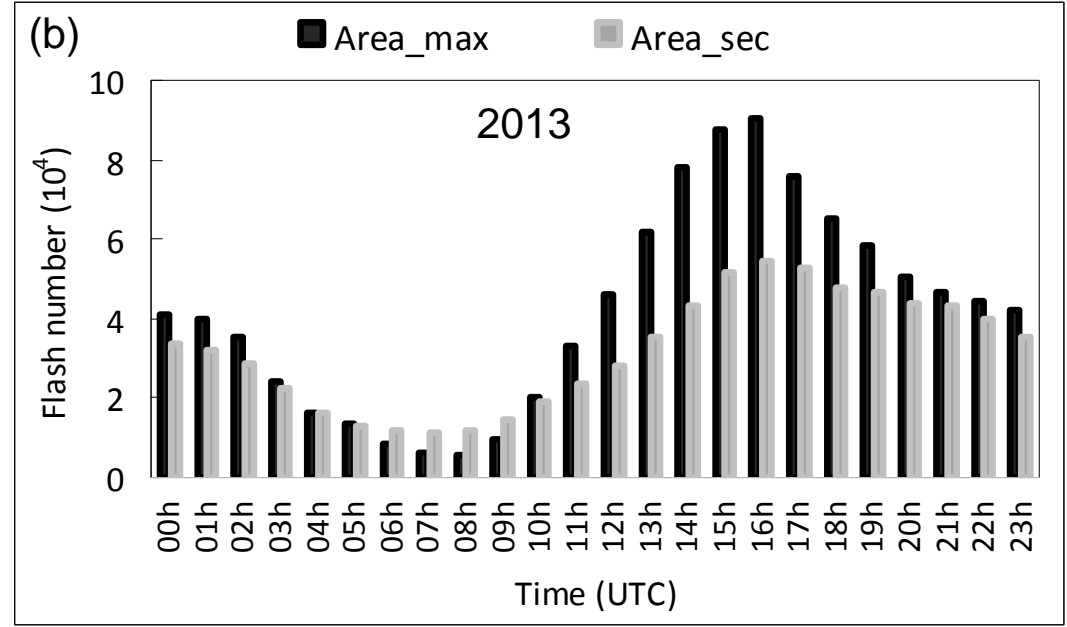

**Figure 3.** Daily evolution of the hourly lightning flash counts in Area_max and Area_sec for 2012 (a) and 2013 (b).

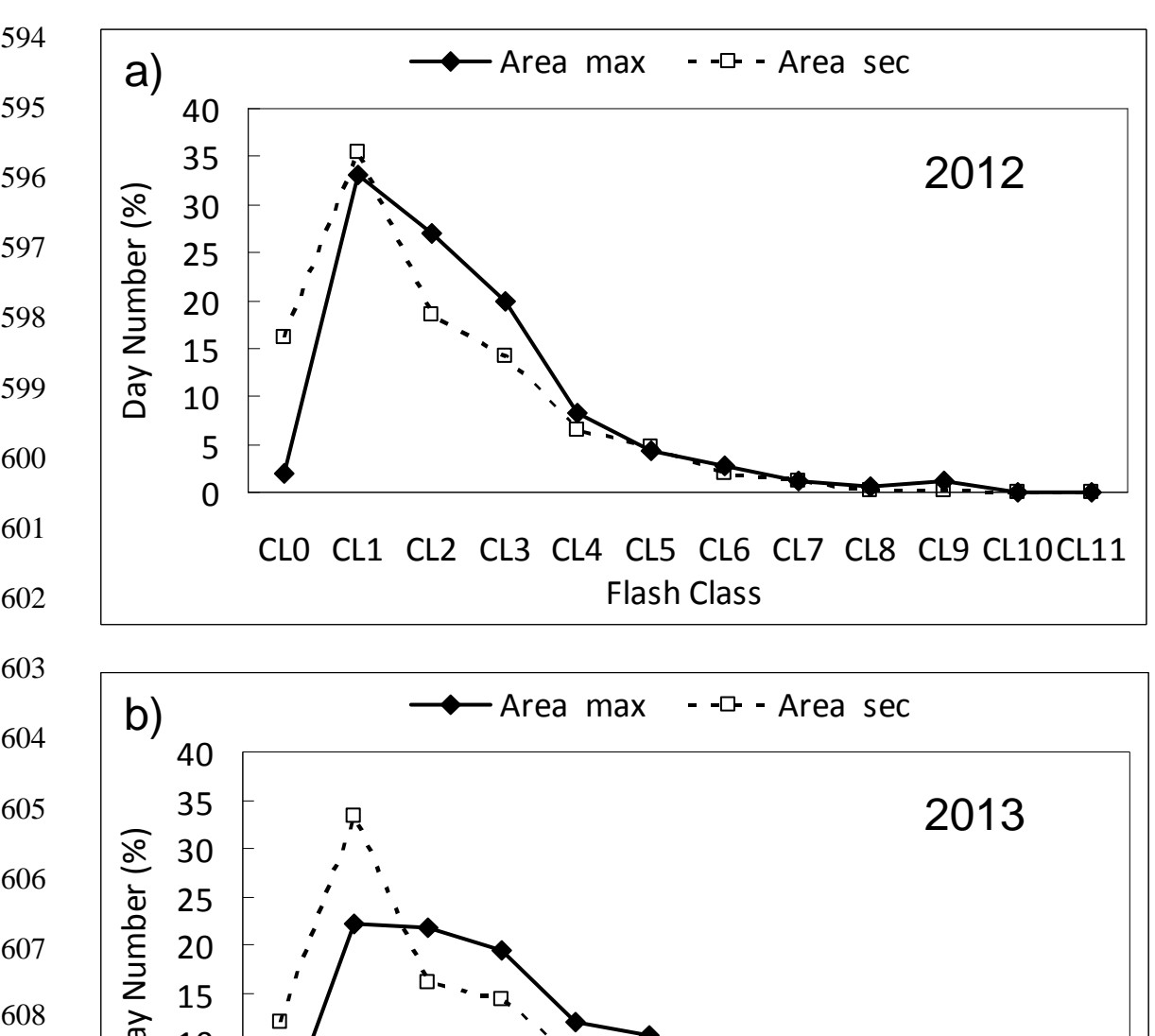

**Figure 4.** Distribution of the number of days (% of the annual number of days) versus the classes of flash number in both areas: (a) for 366 days in 2012, (b) for 362 days in 2013.

619

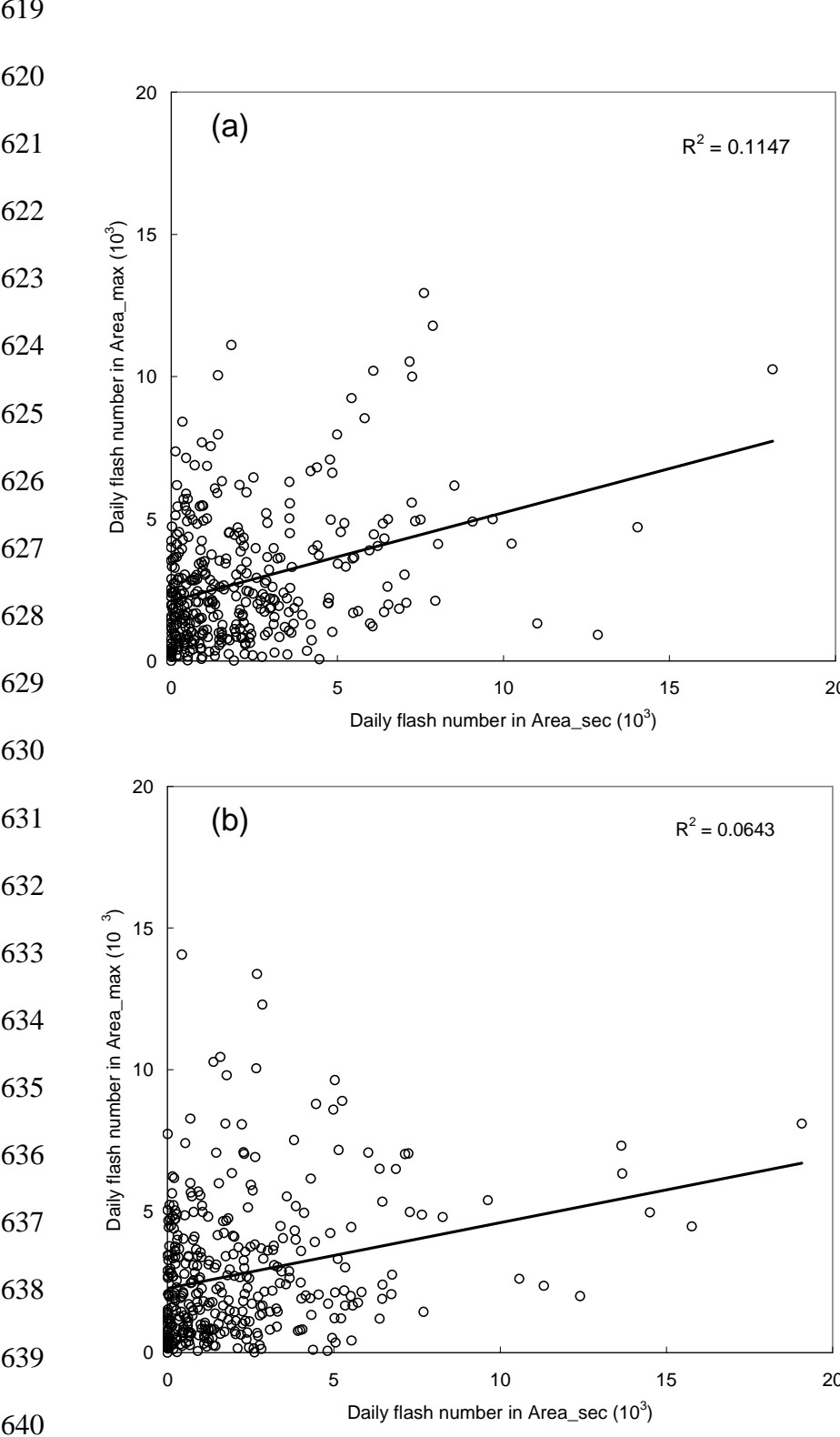

**Figure 5.** Diagrams of correlation between daily numbers of lightning flashes for Area_max and Area_sec in 2013: (a) at calendar daily scale (00h00-24h00 UTC) and (b) at lightning activity daily scale (06h00-06h00 UTC).

644

645

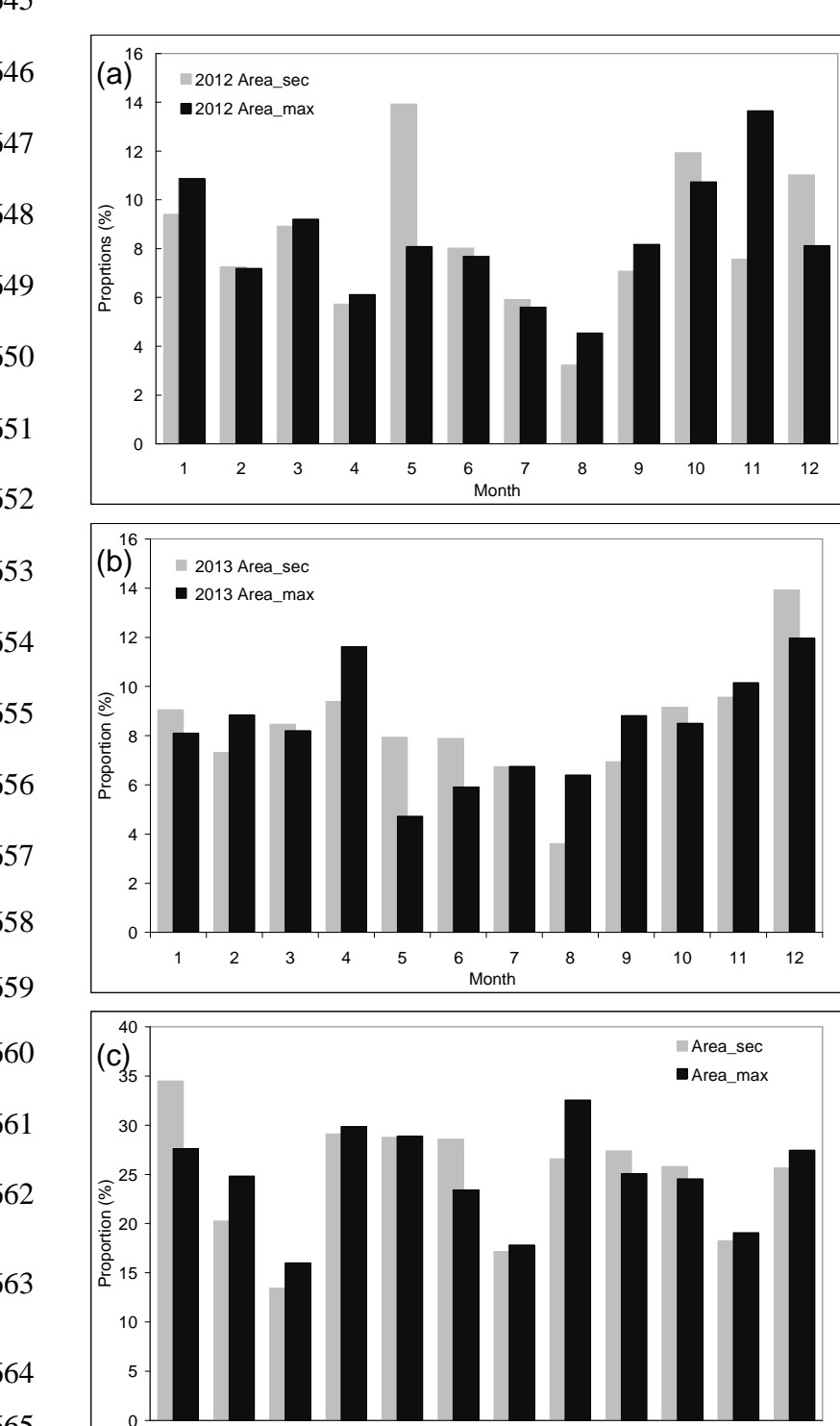

Figure 6. Proportions of flashes detected by WWLLN in Area_max and Area_sec: monthly (a) in 2012 and (b) 2013, and (c) seasonally in the period 2011-2013.





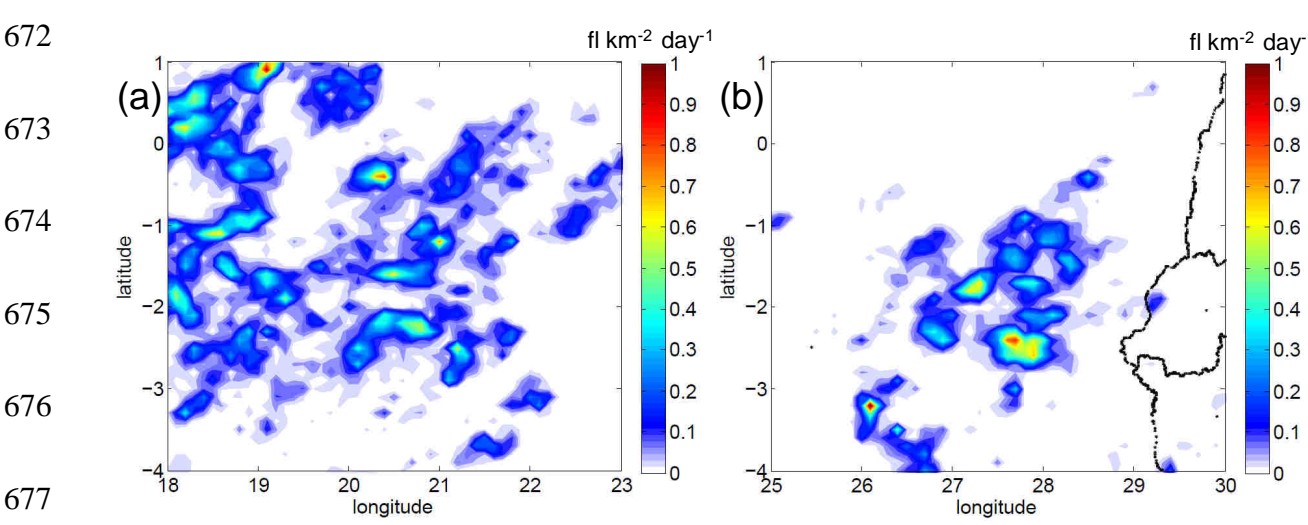






**Figure 7.** Density of lightning flashes (fl km$^{-2}$ day$^{-1}$) detected by WWLLN on 25[th] of December 2013, (a) in Area_sec and (b) in Area_max.


