# Peer review of "Comparison of lightning activity in the two most active areas of the Congo Basin"

_Natural Hazards and Earth System Sciences, 2017_

## Referee Comment (RC1) · Anonymous Referee #1 · 4 Sep 2017

Review for NHESS-2017-105

Title: Comparison of lightning activity in the two 1 most active areas of the Congo Basin
Authors: Kigotsi et al.

General comments: This manuscript presents an exploratory analysis of lightning activity over two distinct areas of Congo Basin: 1) the area where the maximum annual lightning flash rate density (FRD) is observed (west of the mountains that delineate the Rift Valley), hereinafter called Area_max, and 2) the area just west of Area_max, where very high but less pronounced FRD is observed, hereinafter called Area_sec. The manuscript is of the interest to the audience of this journal but needs a few adjustments. I recommend its acceptance only after addressing the issues described below.

[Figure]

Major remarks:

Data: a) Soula et al. (2016) did an excellent job in calculating WWLLN detection efficiency (DE) for each year (2005-2013). This work should leverage from Soula's work and correct 2012 (DE=4.44%) and 2013 (DE=5.90%) data before doing the analysis. The subtle differences from 2012 to 2012 shown here could be an artifact of the different DE. b) Also, why is it relevant to compare 2013 to 2012? Also, was there something different in terms of atmospheric conditions (such as significant droughts, rainier year, El Nino, La Nina, etc.)? My suggestion is to make it simple and combine the years, you may be inserting a lot of uncertainties in your analysis.

Session 3.3: c) I really don't think that the analysis of number of days within classes of flash counts is considered an "Annual variability". d) Also, why use only 2013? e) L 146-147: "The number of days without any flash (CL0) is much larger for Area_sec than for Area_max (7 and 0, respectively).". A difference of only 7 days is not representative of annual variability.

Session 3.4: f) In essence, Fig.3 and Fig. 4 show the same results. Also, the results presented are really confusing making me not to get the relevance of this session.

Session 3.5: g) Did your really expect a correlation between daily number of flashes in each area? This is a very weak way to show that thunderstorms are different within each area and you should rethink how to approach this issue.

Session 3.6: h) Very confusing. . . first of all, "monthly proportions" to what? To total number of lightning in each year? If the objective is to show "monthly activity", why not show flash counts by months? Or is it also the objective to show seasonal contrasts? Please explain better. i) Again, what is the relevance of comparing 2012 to 2013?

Minor remarks:

In general, review the significant figures (or digits) of all your numbers. E.g.: - L 99: ratios of 1.941 and 2.585, shoulb have only one significant digit - L 106: 15.33 flashes

km-2 <yr-1>, should have no significant digit after "point", while 8.22 and 8.62 should be 8.2 and 8.6 (considering that lightning strokes are a single unit)

L 9-23: Avoid using abbreviations in the Abstract text, such as Area_max and Area_Sec, except if explicitly explained in the Abstract.

L 28-29: As a reference, Albrecht et al. (2016) show the impact of resolution (0.1o, 0.25o, 0.5o) while ranking the lightning hotspots. Please see Table ES4 of supplemental material: https://doi.org/10.1175/BAMS-D-14-00193.2.

L 50-52: Table ES4 of supplemental material (https://doi.org/10.1175/BAMS-D-14-00193.2) also shows the persistence of DRC as the second Earth's lightning hotspot.

L 69-88: Please, make it clear that WWLLN detects only cloud-to-ground (CG) lightning and that it does not detect intracloud (IC) lightning, which, in general, is the majority of lightning produced by a thunderstorm. This is also one of the reasons why your values in Fig. 1a differ from those of Albrecht et al. (2016).

L 91, Figure 1: Although your analysis considers full years, the most adequate unit is "flash km-2 yr-1", and it should be called "flash rate density".

L 93: ". . . days of year with thunderstorm activity. . .". Since WWLLN detects CG lightning only, you should substitute "thunderstorm activity" by "lightning activity".

L 98-99: "On the contrary, the flash <rate> density  is very different . . .." . Is that correct?

L 104-105: "By comparing with the values reported by Soula et al. (2016) for a resolution of 0.1°, . . ." which are???

L 115-116: Please give scientific references for this affirmation, or you should state that this is a speculative affirmation.

L 127: "Both areas exhibit the same type of <diurnal lightning activity> evolution with a large. . ."

L130: Please annotate that Local Standard Time (or Solar Time) is the same as UTC (i.e., LST = UTC -0)

L 137-154: You should show only Figure 3 or Table 2, they are redundant. The same is valid for Figure 4 and Table 3.

L160-161: Please define the specific day (or months) regarding the 179 and 92 days span.

L 189-190: "This observation is consistent with the fact that the lightning activity is more spread during the day in Area_sec as indicated in Figure 2.". This may be due to the contribution of nocturnal lightning by MCSs or isolated storms that develop later in the afternoon if compared to Area_max. If you take a closer look in Albrecht et al. (2016) Figure 3, you will see that there is more lightning during the night for the hotspots that are in Area_sec (i.e., 6th and 7th Africa's hotspots).

L 219: "... different locations of our areas". Not really. The daily cycles shown in Albrecht et al. (2016) consider a 1 degree box around the hotspots, and 6 out of 10 Africa's hotspots are within your Area_max and 2 hotspots (Africa's 6th and 7th positions) are within your Area_sec (vide Albrecht's Figs. 2 and 3).

L 219-220: "They found also a more pronounced daily cycle...". This is because they considered a smaller area (a 1 degree box around the hotspots).

Tables 2 and 3: "Number of days", plural in the first line of the tables.

Table 3: explain what (%) means, i.e., proportion to what? The sum of % Is 100% in each column?

Figure 4: Explain "proportaion of day"

[Figure]

---

## Referee Comment (RC2) · E. Williams (Referee) · 12 Sep 2017

This paper takes a look at lightning activity in the "Dark Continent" that also happens to be (often) the leading contributor to global lightning. Accordingly, shedding light on the darkness is a valuable endeavor, and eventually this paper deserves to be published. The most important single need is to identify up front the reasons for the analyses selected, and then to make more detailed physical interpretation of what emerges from the analysis. Several additional areas are identified where improvements can be made below. These substantive issues are followed by detailed comments on the text.

Summary: Consider for publication after major revision

Substantive Issues:

(1) WWLLN documentation

The WWLLN information is the mainstay of this study. Accordingly, more details about WWLLN are needed in the context of the two years selected for study. If differences are documented in selected parameters (Table 1), one would like to know how much of the differenced comes from the detection system and how much is real interannual variability. (That influences the physical interpretation.) So information on the number of receiving stations operating in both years during the period of interest would be helpful. It is widely believed that Africa is generally in third place in the ranking of tropical lightning "chimneys" and that is simply because WWLLN has rather few receiving stations in that part of the world. (In contrast, the other global VLF network, GLD360, is getting Africa much more prominently, but unfortunately Vaisala keeps its information about station numbers and locations secret.) See additional info on this aspect in Williams and Mareev (Atmos. Res., 2009). And toward justifying the scale for gridding of the data, estimates of the accuracy of stroke location are also appropriate. A mention of "continuous increase in detection efficiency" appears in line 77 but without further details. In the Franklin Lecture on "Lightning and Climate" (2012, AGU website), Williams has addressed the problem of the changing detection efficiency in using WWLLN and GLD360 observations as a diagnostic for climate change.

(2) Surface temperature documentation

In other studies, tropical lightning activity has been shown to vary with surface air temperature, also related to CAPE (instability). D. Romps has also shown recently that tropical CAPE may be scaling with the Clausius-Clapeyron relationship, and so there one has a predicted temperature dependence of CAPE. The reviewer has already made inquiry with the second author about this thermodynamic aspect, but the same question is appropriate here. Are surface meteorological observations available at any location in the DRC and in particular for the two areas targeted in this study? That would be a most welcome addition to the physical analysis and interpretation in this paper. The authors need to consider that virtually no additional information is provided

about the surface conditions of the two areas they have selected for study.

(3) Expectations for seasonal variations

The semiannual variation of temperature, rainfall and lightning activity in the climatology of the Congo is well recognized (Christian et al., 2003; Williams and Satori, 2004) but is not mentioned in the interpretation of the Figure 6. In a single year of lightning observations with a detection system that is decidedly inefficient, the semiannual variation may not be so robust, but there are hints of this in Figure 6 already. For example, note the maxima in April in Figure 6b and the local maxima in Figure 6a for October. Also, since the two selected areas are displaced south of the equator, one expects to have an annual phase with maximum in NH winter, also consistent with Figure 6. Place what has been found for localized areas in the broader context of knowledge about Africa.

(4) Positive correlation between lightning areas

My understanding of developments in the Congo is that often convection in the elevated terrain on the eastern boundary develops cold outflows that then go out to the west to stimulate/initiate new convection there. This could be a basis for correlation. Ground conditions are cited, but better would be to cite antecedent rainfall conditions over a large domain that will influence the nature of the convection on subsequent days. I would also strongly recommend another correlation calculation with an area that is immediately adjacent to the primary area, as presently the two selected areas are separate. It would be helpful to show that you have greater correlation when an area immediately adjacent is analyzed.

(5) Role of Lake Kivu

The lake effect is mentioned only briefly (lines 120 and 249) and may deserve some expansion. It is now known that Lake Victoria in Uganda (near to the region of interest) and Lake Maracaibo in Venezuela have dramatic effects on lightning activity. (See

for example the recent work by Albrecht et al. on tropical lightning hot spots, already mentioned.) So more should be said about the physical role of this lake, with possible inclusion of information on its size and about other studies of that role.

(6) The MCS issue

In the last paragraph of the Discussion section, the contribution of mesoscale convective systems is invoked. My big problem with this suggestion is that the authors have already documented the traditional 4 pm maxima in the lightning activity, and that is strongly suggestive of local (solar-stimulated) convection (assisted by cold outflow boundaries) rather than MCS activity that generally maximizes later in the diurnal cycle (and hence the greater prevalence of sprites later in the diurnal cycle, about which the second author is well aware, plus the fact that Africa is the leading "chimney" for sprite activity globally according to ISUAL satellite observations). So I am inclined to agree with what is stated in line 279. But expanded discussion on this aspect is needed. The authors should also consult TRMM work by Karen Mohr on African convection. And given that Zipser et al. (2006) is invoked, the diurnal phase of superlative activity in that study should also be examined and reported here.

(7) Observations with little if any interpretation

The paper has many analyses and observations that are not accompanied by physical interpretation. The Abstract for example contains no physical interpretations at all. Table 3, Figures 3, 4 and 5 are in a similar category. This aspect needs major improvement. It is helpful if every proposed analysis has a specific scientific purpose, and so also warrants an interpretation.

Detailed comments/edits on the text:

The authors are not native English speakers and so there are many edits needed to clean up the text:

Line 18 Suggest dropping "very"

Line 20 "days"

Line 27 Suggest adding Williams and Satori (2004)

Line 28 suggest changing "space" to "spatial"

Line 30 change "instance" to "example"; delete "the"; "from the Lightning. . ."

Line 31 "resolution"

Line 32 "larger dynamic"? Meaning?

Line 34 "maxima"

Line 35 change "both" to "neither"

Line 36 change to "maxima remains throughout the year in considering the lightning activity with 3-month seasons"

Line 37 what is physical interpretation of "very sharp and localized maximum"

Line 38 "in the eastern Democratic. . ."

Line 42 "scattered over a large area"

Line 43 "maximum activity could. . ."

Line 43 "linear scale for flash density was. . ."

Line 46 "maximum activity"

Line 48 change "whole" to "entire"

Line 49 "most of them quantified"

Line 51 "maximum in flash density"

Line 53 "The geographical extent of this region"

Line 57 "high spatial resolution"; "allowed a better localization and specification of its shape"

Line 62 "contrasting from year to year"

Line 63 "extends roughly"

Line 66 "maximum activity"

Line 70 change "dimension" to "area"

Line 77 Attributable to what? (see earlier discussion)

Line 78 "the last two years"

Line 81 "radiation"

Line 84 delete the first "the"

Line 85 quantify "very little attenuation"; it is not small and for this reason large numbers of sensors are needed for global surveillance

Line 87 Why report this for 2014 when it is 2012 and 2013 that are used for analysis?

line 91 This would be 5 km resolution. You should justify that in terms of the accuracy of the stroke location in Africa.

line 94 "with the same"

Line 95 "the flash count"

Line 96 "the maximum flash density for both areas and for each year"

Line 97 "exhibit total flash counts"

Line 97-98 "indicates a stable situation from one year to the next. In contrast, the ratio..."

Line 99 "one year to the next" 4 digits here is overkill on precision

Line 101 "localized"; "one year to the next"; "Furthermore, the spatial density..."

Line 103 "depends on the spatial resolution"

Line 104 "at a resolution"

Line 105 "maximum of flash density"

Line 109 "clearly appears"

Line 114 "thunderstorms, which means that the number of flashes per day is larger..."

Lines 115-116 These two factors could be distinguished with WWLLN observations but you need to check the temporal development.

Line 117 I hope the authors disclose "specific and local conditions"

Line 119 Which side and why?

Line 120 "increases markedly"

Line 124 "daily cycle of flashes detected by the WWLLN"

Line 125 "These flash counts are calculated..."

Line 126 "so that the flashes are associated with the ..."

Line 129 "for the minima in the morning..."

Line 130 "and for the maxima in the afternoon..."

Line 131 "contrast in flash counts between..."

Line 134 Add comma after "day"

Lines 133 to 136 What is your interpretation?

Line 138 "distribution of flashes"

Line 139 "year of reference"? Only one year?

Line 142 How were the various classes selected?

Line 145 "also plotted in Figure 3" (reduce redundancy)

Line 148 "number of days"; "about twice that of Area..."; "157 versus 84"

Line 155 "Variability of flash counts during..."

Line 157 "a clear minimum activity"

Line 160 "defined as the high activity" But you haven't quantified HAP and LAP.

Line 165 change to "and also in roughly the same proportion..."

Line 166 "with number of flashes exceeding 5000 (CL6-CL111)"

Line 167 "during the LAP"

Line 169 "during the HAP and the LAP"

Line 170 "During the HAP"

Line 171 "of days"

Line 172 "number less than 5000"; "whereas during the LAP"

Line 174 You don't have a real motivation here. Tell why you might expect correlated behavior.

Line 178 For this you should be giving local times, not UT times. Otherwise you lose

the physical interpretation.

Line 180 You should be reporting correlation coefficients in the text in the same form as in the figures. Otherwise this is potentially confusing.

Line 182 "it also increases for the other"; "first glance" of what?

Line 190 "is more widely distributed during the day"

Line 192 I don't understand the meaning here? Clear-cut?

Line 193 Shouldn't this section be merged with 3.3 Annual variability. It is the same topic.

No discussion of the important semiannual variation in this section.

Line 194 "proportion"

Line 206 suggest adding text: "based on satellite optical observations of lightning" to distinguish from the approach taken here with VLF data. You should also define "hotspot"

Lines 210-211 What did A. Laing say in there about MCSs?

Line 214 Considered by whom? These are not the times considered in Section 3.2.

Line 216 This is yet another time interval.

Line 217 This is not what you reported in lines 128-130.

Line 219 "locations than our two areas"

Line 224 "result for 2011" on WWLLN ? Please clarify.

Line 225 What is the meaning of "minimum proportion"?

Line 228 The authors need to articulate their views on the ITCZ in the lightning context. In my experience, the activity lightning is usually adjacent to the ITCZ because one needs subsidence to eliminate the widespread cloudiness that is shuts off the destabilizing influence of sunlight.

236-240 Nothing is included in here about antecedent conditions of rainfall, that can influence the Bowen ratio. See also Williams and Stanfill (2002; Comptes Rendues).

Line 249 Need more discussion on the role of "great lakes" in the lightning context

Line 250 "for the development"

Line 251 "at the planetary scale"; when do "the most intense storms" max out in the diurnal cycle? Are they isolated, or are they parts of MCSs?

Line 256 "spread from the east to the western Congo basin"

Line 257 Only if MCS status. But don't forget role of cold outflow toward the west.

Line 259 And antecedent rainfall. In any case, more should be said about the nature of the surface in the areas selected. In this context, Williams and Satori (2004) should be consulted.

Line 263 "regions of strong coupling between the atmosphere. . ."

Lines 264-265 One does not want a contrast if one is seeking to explain correlated behavior.

Line 266 "mesoscale convective systems"

Line 269 "in the Congo basin"

Line 270

"frequently overshoots the tropopause. The climatology. . ."

Line 272 "From a five-year series of data..."

Line 273 "to the western side of the high mountains"

Line 275 "maxima in the number"

Line 279 I tend to agree with this statement but the discussion on MCSs needs to be elaborated on here.

Conclusions, like Abstract, is lacking in physical interpretation.

Line 282 "The spatial and temporal characteristics of the lightning..."

Line 282 "strongest thunderstorm activity"

Line 283-284 change to "with a secondary maximum"; "concentrated in the same part"

Line 287 to 288 "is similar in both areas

References

Suggest adding Williams and Satori (2004)

Williams et al. (2000, JAM) considers variations of tropical flash rates and diurnal cycles of flash rates and storm counts.

Williams (2012, Franklin Lecture) considers impact of changes in WWLLN detection efficiency over time.

Table 1 Two significant figures is probably more appropriate. In some places the authors use four!

Figure 1 The hotspot areas straddle the equator. Some discussion is needed about that aspect alone in driving the lightning counts up high. These zones are visited at least twice per year by the zone of instability. Caption could also mention location accuracy of individual strokes.

Figure 2 Suggest changing "amounts" to "counts" Please compare this variation with

those documented in Williams et al. (JAM, 2000). 4 pm is very consistent, and with Schumann resonance observations of "background"

Figure 4 Better to show flash counts that CLi classes, which require going elsewhere to check on definition/motivation. What is the thermodynamic situation on days with > CL10? Curious minds want to know.

Figure 5 If R^2 value are used here, same values should be discussed in the text.

Figure 6 Need more discussion on semiannual and annual variations in general. (See earlier remarks.)

---

## Author Comment (AC1) · 16 Oct 2017

Response to Anonymous Referee #1

Review for NHESS-2017-105

Title: Comparison of lightning activity in the two 1 most active areas of the Congo Basin Authors: Kigotsi et al.

General comments:

This manuscript presents an exploratory analysis of lightning activity over two distinct

areas of Congo Basin: 1) the area where the maximum annual lightning flash rate density (FRD) is observed (west of the mountains that delineate the Rift Valley), hereinafter called Area_max, and 2) the area just west of Area_max, where very high but less pronounced FRD is observed, hereinafter called Area_sec. The manuscript is of the interest to the audience of this journal but needs a few adjustments. I recommend its acceptance only after addressing the issues described below.

Response of the authors

The authors thank the reviewer for her/his careful work to evaluate the paper. We appreciate the comments and the remarks that help to improve the paper. The paper required a major revision and we hope to have made corrections enough to make the paper clearer and more relevant paper.

Substantial modifications are made, especially a figure is added to have a wider view of the data and justify some choices. The study is systematically extended to 2012 data, to have a more robust comparison between both areas, which is the goal of the paper. We delete a figure and add a new graph and a new figure to show one case of distribution of a strong daily lightning activity. We add information about the WWLLN data and network.

The interpretation is developed when possible. For example we now highlight an interpretation for the difference between both areas by using the paper by Jackson about MCS location over equatorial Africa: both areas (Area-max and Area_sec) are included in one of the four maximums described in Jackson et al.. They explain this large maximum is due to the AEJ-S, while two other maximums were explained by the orography and another by the Lake Victoria. We distinguish two maximums in this large maximum, from which Area_max combines the presence of AEJ-S with local orography and Lake Kivu.

We make most of the corrections suggested by the reviewers and we answer to the comments in the following.

[Figure]

Major remarks:

Data: a) Soula et al. (2016) did an excellent job in calculating WWLLN detection efficiency (DE) for each year (2005-2013). This work should leverage from Soula's work and correct 2012 (DE=4.44%) and 2013 (DE=5.90%) data before doing the analysis. The subtle differences from 2012 to 2012 shown here could be an artifact of the different DE.

Response of the authors:

First of all, we have to say the comparison is made between two areas with a large flash rate density (FRD) in Congo Basin and not from one year to the next.

These areas (Area_max and Area_sec) correspond to the maximums pointed out in Soula et al. (2016) and as the reviewer noted it in a comment, to the areas surrounding most hotspots in Africa noted by Albrecht et al. (2016). Area_max includes 6 out of the 10 hotspots (1,2,3,5,8 and 10) found in Albrecht et al., while Area_sec includes 2 out of the 10 hotspots (6 and 7).

The DE is considered in Soula et al. (2016) and it was calculated relatively to the LIS data that cumulate cloud-to-ground and intracloud flashes. Thus, the DE values found in Soula et al. are low for the whole study area, 5.9% and 4.4% for 2013 and 2012, respectively. However, the DE can depend on the region since the study area in Soula et al. was very large ($25° \times 25°$). Soula et al. (2016) have clearly highlighted the increase of DE between 2012 and 2013, the rate of which can be estimated at about 34%.

We noted also the DE was not constant in the whole study area considered in Soula et al. (2016). Thus, the values 4.44% and 5.90% are average values for the whole area. We consider now the specific values of DE for both areas Area_max and Area_sec. The new figure 1 is made to show different parameters for each area from 2005 to 2013:

- the lightning activity issued from LIS

- the lightning activity issued from WWLLN

- the DE estimation calculated according to the methodology presented in Soula et al.

We see DE was stronger in Area_sec from 2005 to 2009 and in Area_max from 2010 to 2013. The question to correct the data by applying DE can be asked. We choose to let the data without any corrections for several reasons:

- the correction can be applied only globally for a given area, it does not change the comparison of the parameters we compare between both areas when we use proportions (proportion of lightning versus month)

- the DE is calculated for one year and for a given area. To take into account an eventual correction we have to add flashes uniformly in each month, in each 1-hour time interval, in each day... It seems too artificial to correct all flash numbers at such small scales as 1-hour window, day, month...

- The correction could be made at the scale of the year for the number of flashes.

b) Also, why is it relevant to compare 2013 to 2012? Also, was there something different in terms of atmospheric conditions (such as significant droughts, rainier year, El Nino, La Nina, etc.)? My suggestion is to make it simple and combine the years, you may be inserting a lot of uncertainties in your analysis.

Response of the authors:

Figure 1 can be a response to the comment because it provides an overview of LIS and WWLLN data over the 9-year period. The two years 2012 and 2013 are selected because they correspond to the strongest detection efficiency (DE) from the years we have in our database.

In Soula et al. (2016) the LIS data were used to compare the activity from one year to the next. The difference for the whole region was low since the maximum was found in

2009 (195,316 flashes detected) and the minimum was found in 2012 (182,560 flashes detected), which provides a difference of 6.5%. Considering 2013, LIS data provides 192,443 flashes detected which represents an increase of about 5% from 2012. The interannual variability was found low by considering LIS data. Now we consider for this work of comparison the DE at the scale of each area (Area-max and Area_sec) and the LIS data at each area too. The new information allows better describing the WWLLN data used in this study.

Session 3.3: c) I really don't think that the analysis of number of days within classes of flash counts is considered an "Annual variability".

Response of the authors:

Done, we use now Day-to-day variability

d) Also, why use only 2013?

Response of the authors:

We use 2012 and 2013 for a study more robust.

e) L146-147: "The number of days without any flash (CL0) is much larger for Area_sec than for Area_max (7 and 0, respectively).". A difference of only 7 days is not representative of annual variability.

Response of the authors:

We change the first class because we now think it is not necessary to separate days without any flash and days with very low flash numbers (some cases have less than 10 flashes). Thus we consider now a first class corresponding to a very low flash rate (< 100 flashes per day in an area).

Session 3.4:

f) In essence, Fig.3 and Fig. 4 show the same results. Also, the results presented are

really confusing making me not to get the relevance of this session.

Response of the authors:

Section 3.4 is deleted.

Session 3.5:

g) Did your really expect a correlation between daily number of flashes in each area? This is a very weak way to show that thunderstorms are different within each area and you should rethink how to approach this issue.

Response of the authors:

We explain at the beginning the approach that consist in comparing the lightning activity day by day. It allows us to show the strong lightning activity is often local, even if the conditions favourable for storm developments are present in larger areas. Figure 7 shows an example of daily lightning flash rate density.

Session 3.6:

h) Very confusing: : : first of all, "monthly proportions" to what? To total number of lightning in each year? If the objective is to show "monthly activity", why not show flash counts by months? Or is it also the objective to show seasonal contrasts? Please explain better.

Response of the authors:

The section aims to present the annual distribution of the lightning activity, at the scale of the month. We call it now "Month-to-month variability". We add a figure to show the annual cycle at the scale of the season defined by DJF, MAM, JJA and SON, as in Christian et al. (2003).

i) Again, what is the relevance of comparing 2012 to 2013?

Response of the authors:

[Figure]

We do not compare 2012 to 2013, the reason for considering two years is to have a more robust comparison between two areas.

Minor remarks:

In general, review the significant figures (or digits) of all your numbers. E.g.: - L 99: ratios of 1.941 and 2.585, shoulb have only one significant digit –

Response of the authors:

We agree and correct. Two digits after dot are fine. 0.01 over 1 is about 1%.

L 106: 15.33 flashes km-2 <yr-1>, should have no significant digit after "point", while 8.22 and 8.62 should be 8.2 and 8.6 (considering that lightning strokes are a single unit)

Response of the authors:

For the values around 8 for the flash density, effectively one digit after dot seems enough because 0.02 over 8 is about 0.25%. Consequently, one digit for 15.33 seems also enough, it would be 15.3.

L 9-23: Avoid using abbreviations in the Abstract text, such as Area_max and Area_Sec, except if explicitly explained in the Abstract.

Response of the authors:

At the beginning of abstract (first sentence), Area_max and Area_sec are explained.

L 28-29: As a reference, Albrecht et al. (2016) show the impact of resolution (0.1o, 0.25o, 0.5o) while ranking the lightning hotspots. Please see Table ES4 of supplemental material: https://doi.org/10.1175/BAMS-D-14-00193.2.

Response of the authors:

Thank you for this comment about the very instructive table. The initial comment we made in the paper was essentially related to the shape of the maximum area in the

Congo basin. We note the reference to illustrate the effect of the spatial resolution on the maximum value of FRD and on its location and we develop the comments related to this aspect.

L 50-52: Table ES4 of supplemental material (https://doi.org/10.1175/BAMS-D-14-00193.2) also shows the persistence of DRC as the second Earth's lightning hotspot.

Response of the authors:

The response in the previous point includes the response to this comment.

L 69-88: Please, make it clear that WWLLN detects only cloud-to-ground (CG) lightning and that it does not detect intracloud (IC) lightning, which, in general, is the majority of lightning produced by a thunderstorm. This is also one of the reasons why your values in Fig. 1a differ from those of Albrecht et al. (2016).

Response of the authors:

We do not compare the values of the FRD in our paper with those in Albrecht et al. (2016) since they are not comparable. However, according to several references the WWLLN can detect IC flash strokes but with a lower detection efficiency. The system does not exclude the IC strokes, which could be made probably with a recognition of form.

For example, Rodgers et al. (2005) say :" The detection efficiency of the WWLL is also considered. In the selected region the WWLL detected _13% of the total lightning, suggesting a 26% CG detection efficiency and a 10% IC detection efficiency."

Abarca et al. (2011) says: "The network detects CG and intracloud (IC) flashes with the same efficiency as long as they have the same current magnitude and channel length (Lay et al. 2004; Rodger et al. 2005, 2006; Jacobson et al. 2006); however, CG DE is about twice the IC DE (Abarca et al. 2010) because CG flashes tend to have higher peak currents."

We note the WWLLN is less efficient for IC flash detection.

Abarca, S.F., Corbosiero, K.L., Vollaro D., 2011. The World Wide Lightning Location Network and convective activity in tropical cyclones. Mon. Weather Rev. 139, 175–191.

Rodger, C.J., Brundell, J.B., Dowden, R.L., 2005. Location accuracy of long distance VLF lightning location network: post algorithm upgrade. Ann. Geophys. 23, 277–290.

L 91, Figure 1: Although your analysis considers full years, the most adequate unit is "flash km-2 yr-1", and it should be called "flash rate density".

Response of the authors:

Done

L 93: ": : : days of year with thunderstorm activity: : :". Since WWLLN detects CG lightning only, you should substitute "thunderstorm activity" by "lightning activity".

Response of the authors:

The WWLLN detects also IC flashes, so thunderstorm activity can be used but lightning activity can be well adapted.

L 98-99: "On the contrary, the flash <rate> density  is very different : : :." . Is that correct?

Response of the authors:

We compare the ratio between the maxima flash (rate) densities in both areas, calculated in 2012 and in 2013 (Table 1). The ratio for one year can be different in one year and in the other.

L 104-105: "By comparing with the values reported by Soula et al. (2016) for a resolution of 0.1_, : : :" which are???

Response of the authors:

The sentence that follows in the text gives these values. Maybe we are not clear, we try to improve it.

L 115-116: Please give scientific references for this affirmation, or you should state that this is a speculative affirmation.

Response of the authors:

It was noted in Soula et al. (2016). We note the number of flashes per stormy day is larger in the region of the main maximum. To have more flashes during a day of storm, there are three possible explanations: more storms, storms more active, storms more stationary. It can be also a combination of several of the three explanations.

L 127: "Both areas exhibit the same type of <diurnal lightning activity> evolution with a large: : :"

Response of the authors:

Done

L130: Please annotate that Local Standard Time (or Solar Time) is the same as UTC (i.e., LST = UTC -0)

Response of the authors:

We note this sentence at the beginning of the section: "The time is indicated in UTC, which is two hours late compared to Local Time (LT = UTC + 2)." Be careful, the local time is different in western DRC and eastern DRC and local time is different from solar time that needs a calculation. Local time is the time used in the eastern part of the country (DRC) including both areas (Area_sec and Area_max).

L 137-154: You should show only Figure 3 or Table 2, they are redundant. The same is valid for Figure 4 and Table 3.

Response of the authors:

Tables are rearranged. Table 3 is deleted and the new table 2 includes now 2012 and 2013 data. Figure 4 is deleted and the new figure 4 includes 2012 (a) and 2013 (b). The table provides the number of days for each class and the percentage of the total number of days. The figure has its usefulness for the tendency of the evolution in each area and their comparison.

L160-161: Please define the specific day (or months) regarding the 179 and 92 days span.

Response of the authors:

Deleted

L 189-190: "This observation is consistent with the fact that the lightning activity is more spread during the day in Area_sec as indicated in Figure 2.". This may be due to the contribution of nocturnal lightning by MCSs or isolated storms that develop later in the afternoon if compared to Area_max. If you take a closer look in Albrecht et al. (2016) Figure 3, you will see that there is more lightning during the night for the hotspots that are in Area_sec (i.e., 6th and 7th Africa's hotspots).

Response of the authors:

Good point. We add this comment: "This may be due to the contribution of nocturnal lightning by MCSs or isolated storms that develop later in the afternoon if compared to Area_max. Indeed, the work by Albrecht et al. (2016) shows in their Figure 3 that during the night, the hotspots located in Area_sec (i.e, 6th and 7th Africa's hotspots) exhibit a larger contribution to the daily lightning activity.

L 219: ": : : different locations of our areas". Not really. The daily cycles shown in Albrecht et al. (2016) consider a 1 degree box around the hotspots, and 6 out of 10 Africa's hotspots are within your Area_max and 2 hotspots (Africa's 6th and 7th positions) are within your Area_sec (vide Albrecht's Figs. 2 and 3).

Response of the authors:

We agree and the sentence did not express correctly what we wanted to say. We say now : "for several hotspots located in our areas"

L 219-220: "They found also a more pronounced daily cycle: : :". This is because they considered a smaller area (a 1 degree box around the hotspots).

Response of the authors:

We change the sentence to say our results are consistent with those from Albrecht et al.

Tables 2 and 3: "Number of days", plural in the first line of the tables.

Response of the authors:

Done

Table 3: explain what (%) means, i.e., proportion to what? The sum of % Is 100% in each column?

Response of the authors:

Deleted

Figure 4: Explain "proportaion of day"

Response of the authors:

Figure 4 is deleted but the proportion is still used. We now explain the proportion of days in the caption of the new figure 4 and in the caption of Table 2.

Please also note the supplement to this comment:
https://www.nat-hazards-earth-syst-sci-discuss.net/nhess-2017-105/nhess-2017-105-AC1-supplement.pdf
* * *
2017-105, 2017.

---

## Author Comment (AC2) · 16 Oct 2017

Response to the Review of "Comparison of lightning activity in the two most active areas of the Congo Basin" by J.K. Kigotsi, S. Soula and J.-F. Georgis

This paper takes a look at lightning activity in the "Dark Continent" that also happens to be (often) the leading contributor to global lightning. Accordingly, shedding light on the darkness is a valuable endeavor, and eventually this paper deserves to be published. The most important single need is to identify up front the reasons for the analyses selected, and then to make more detailed physical interpretation of what emerges from the analysis. Several additional areas are identified where improvements can be made below. These substantive issues are followed by detailed comments on the text.

Summary: Consider for publication after major revision

Response of the authors

The authors thank Earle Williams for his detailed and useful work to evaluate the paper. We appreciate the comments and the remarks that help to improve the paper. The paper required a major revision and we hope to make corrections enough to obtain a clearer and more relevant paper.

Substantial modifications are made, especially a figure is added to have a wider and more precise view of the data and justify some choices. We add information about the WWLLN data and network. The study is also extended to 2012 data, to have more robust results from the comparison between both areas, which is the goal of the paper. We delete a figure and add a new graph and a new figure to show one case of lightning distribution during a strong daily lightning activity. The interpretation is developed when possible. For example we now highlight an interpretation for the difference between both areas by using the paper by Jackson about MCS location over equatorial Africa: both areas (Area-max and Area_sec) are included in one of the four maximums described in Jackson et al.. They explain this large maximum is due to the AEJ-S, while two other maximums were explained by the orography and another by the Lake Victoria. We distinguish two maximums in this large maximum, from which Area_max combines the presence of AEJ-S with local orography and Lake Kivu.

We make most of the corrections suggested by the reviewers and we answer to the comments in the following.

Substantive Issues:

(1) WWLLN documentation

The WWLLN information is the mainstay of this study. Accordingly, more details about WWLLN are needed in the context of the two years selected for study. If differences are documented in selected parameters (Table 1), one would like to know how much of the

differenced comes from the detection system and how much is real interannual variability. (That influences the physical interpretation.) So information on the number of receiving stations operating in both years during the period of interest would be helpful. It is widely believed that Africa is generally in third place in the ranking of tropical lightning "chimneys" and that is simply because WWLLN has rather few receiving stations in that part of the world. (In contrast, the other global VLF network, GLD360, is getting Africa much more prominently, but unfortunately Vaisala keeps its information about station numbers and locations secret.) See additional info on this aspect in Williams and Mareev (Atmos. Res., 2009). And toward justifying the scale for gridding of the data, estimates of the accuracy of stroke location are also appropriate. A mention of "continuous increase in detection efficiency" appears in line 77 but without further details. In the Franklin Lecture on "Lightning and Climate" (2012, AGU website), Williams has addressed the problem of the changing detection efficiency in using WWLLN and GLD360 observations as a diagnostic for climate change.

Response of the authors:

First of all, we have to say the comparison is made between two areas with a large flash rate density (FRD) in the Congo basin. These areas (Area_max and Area_sec) correspond to the maximums pointed out in Soula et al. (2016) and to the areas surrounding the hotspots in Africa noted by Albrecht et al. (2016). Area_max includes 6 out of the 10 hotspots (1,2,3,5,8 and 10) found in Albrecht et al., while Area_sec includes 2 out of the 10 hotspots (6 and 7). The comparison is not made from one year to another. The two years 2012 and 2013 are selected because they correspond to the strongest detection efficiency (DE) from the years we have in our database. The DE is considered in Soula et al. (2016) and it was calculated relatively to the LIS data that cumulate cloud-to-ground and intracloud flashes. Thus, the DE values found in Soula et al. are low for the whole study area, 5.9% and 4.4% for 2013 and 2012, respectively. However, the DE can depend on the region since the study area in Soula et al. was very large ($25° \times 25°$). It is difficult to have a report on the WWLLN status during these

two years. Anyway, Soula et al. (2016) have clearly highlighted the increase of DE between 2012 and 2013, the rate of which can be estimated at about 34%.

In Soula et al. (2016) the LIS data were used to compare the activity from one year to the next. The difference for the whole region was low since the maximum was found in 2009 (195,316 flashes detected) and the minimum was found in 2012 (182,560 flashes detected), which provides a difference of 6.5%. Considering 2013, LIS data provides 192,443 flashes detected which represents an increase of about 5% from 2012. The interannual variability was found low by considering LIS data. Now, for this comparison study, we consider the DE at the scale of each area (Area-max and Area_sec) and the LIS data are used in each area (see Figure 1 in the new version of the paper). The new information allows better describing the WWLLN data used in this study. A new graph in Figure 1 displays the annual count of lightning flashes from LIS and WWLLN for each area and during the whole data period (2005-2013), and the DE values calculated in each area with the method used by Soula et al. (2016). The years 2013 and 2012 have the larger values of DE, which can justify to take these two years of reference for the comparison between both Area_max and Area_sec.

(2) Surface temperature documentation

In other studies, tropical lightning activity has been shown to vary with surface air temperature, also related to CAPE (instability). D. Romps has also shown recently that tropical CAPE may be scaling with the Clausius-Clapeyron relationship, and so there one has a predicted temperature dependence of CAPE. The reviewer has already made inquiry with the second author about this thermodynamic aspect, but the same question is appropriate here. Are surface meteorological observations available at any location in the DRC and in particular for the two areas targeted in this study? That would be a most welcome addition to the physical analysis and interpretation in this paper. The authors need to consider that virtually no additional information is provided about the surface conditions of the two areas they have selected for study.

Response of the authors:

It is difficult to find surface temperature data in the region considered in the study. Anyway, to use the temperature in the study requires the knowledge of its values in several locations of the areas and the consideration of the altitude. We look for differences of the storm characteristics between two regions. The characteristics investigated are, the daily cycle, the distribution of the FRD, the annual cycle month by month and season by season, the distribution of the number of flashes produced during a day. The highlighting of differences has to be interpreted in physical concept with the available information.

(3) Expectations for seasonal variations

The semiannual variation of temperature, rainfall and lightning activity in the climatology of the Congo is well recognized (Christian et al., 2003; Williams and Satori, 2004) but is not mentioned in the interpretation of the Figure 6. In a single year of lightning observations with a detection system that is decidedly inefficient, the semiannual variation may not be so robust, but there are hints of this in Figure 6 already. For example, note the maxima in April in Figure 6b and the local maxima in Figure 6a for October. Also, since the two selected areas are displaced south of the equator, one expects to have an annual phase with maximum in NH winter, also consistent with Figure 6. Place what has been found for localized areas in the broader context of knowledge about Africa.

Response of the authors:

In Soula et al. (2016) the DE values for the WWLLN were already noted as low and discussed. First, we have to keep in mind the DE is calculated relatively to the LIS sensor that detects all flashes (intracloud and cloud-to-ground). Since the WWLLN detects principally the CG flashes (but also some IC flashes - see references added in the paper as Abarca et al., 2011 and Rodgers et al., 2005), the values of DE are obviously low. Thus, the DE values are indicative and what is interesting is to follow the

DE values year after year. A major result found in Soula et al. is that according to the high flash rate within this region, a low proportion of flashes detected is representative of the climatology. It is true also in the present study.

Concerning the semiannual variability of the monthly rate, it has been found in Soula et al. it can be large at the scale of the Congo basin. Consequently, the variability is large for the restricted areas considered in the present study. It is of course arbitrary to consider month by month to analyze the semiannual variability of the lightning activity. It is also possible to consider the 3-month averaged flash proportion to smooth the effect of a specific month. The reference to choose the 3-month period is based on typical periods considered in other studies in the region (Christian et al., 2003; Jackson et al., 2009) and from Soula et al. (2016) that made an average annual cycle from 9 years. Thus, the four periods are DJF, MAM, JJA and SON. We add this approach in the new figure 6 to point out the semiannual variation. It is now commented for a complete and continuous cycle of three years. We discuss the result for this figure in the context of the knowledge about Africa.

(4) Positive correlation between lightning areas

My understanding of developments in the Congo is that often convection in the elevated terrain on the eastern boundary develops cold outflows that then go out to the west to stimulate/initiate new convection there. This could be a basis for correlation. Ground conditions are cited, but better would be to cite antecedent rainfall conditions over a large domain that will influence the nature of the convection on subsequent days. I would also strongly recommend another correlation calculation with an area that is immediately adjacent to the primary area, as presently the two selected areas are separate. It would be helpful to show that you have greater correlation when an area immediately adjacent is analyzed.

Response of the authors:

This correlation study was made to check if the days with activity can correspond between two areas, in such a way that the conditions favorable for storms could affect both areas. We have chosen the areas because they correspond to the study. The result is a weak correlation. We have to keep in mind the correlation is evaluated between the daily numbers of flashes in each area. It is a quantitative correlation. After analyzing some case studies in this region, we can see the strong flash rate density is always very localized in a restricted zone, that is to say the strong activity is not extended. It can explain or help to understand the weak correlation between the flash numbers in each $5° \times 5°$ area. At the scale of one day the large flash rate/density (and strong rainfall for example) is local. We explain that and we show an example of day with strong activity in both areas (Figure 7). It shows also that even if the areas are adjacent, the correlation can be weak.

Role of Lake Kivu

The lake effect is mentioned only briefly (lines 120 and 249) and may deserve some expansion. It is now known that Lake Victoria in Uganda (near to the region of interest) and Lake Maracaibo in Venezuela have dramatic effects on lightning activity. (See for example the recent work by Albrecht et al. on tropical lightning hot spots, already mentioned.) So more should be said about the physical role of this lake, with possible inclusion of information on its size and about other studies of that role.

Response of the authors:

We have discussed the possible effect of the lakes in Soula et al. (2016). We add some comments. We can use the figure 2 to show the effect of Lake Victoria, especially on the number of days. We can see a clear enhancement above the lake. This enhancement is less visible for the lightning flash rate density. It means the number of flashes per day of storm is lower than in other parts of the area. The storms are therefore frequent above the lake Victoria, but not very active in terms of lightning flash production.

(5) The MCS issue

In the last paragraph of the Discussion section, the contribution of mesoscale convective systems is invoked. My big problem with this suggestion is that the authors have already documented the traditional 4 pm maxima in the lightning activity, and that is strongly suggestive of local (solar-stimulated) convection (assisted by cold outflow boundaries) rather than MCS activity that generally maximizes later in the diurnal cycle (and hence the greater prevalence of sprites later in the diurnal cycle, about which the second author is well aware, plus the fact that Africa is the leading "chimney" for sprite activity globally according to ISUAL satellite observations). So I am inclined to agree with what is stated in line 279. But expanded discussion on this aspect is needed. The authors should also consult TRMM work by Karen Mohr on African convection. And given that Zipser et al. (2006) is invoked, the diurnal phase of superlative activity in that study should also be examined and reported here.

Response of the authors:

We agree with the reviewer that the reference Zipser has to be more commented. The work is supposed to point out differences between two areas and the diurnal cycle appears different in both areas. Since it is more pronounced in Area_max it indicates more activity issued from local conditions, what is said (presence of mountains, lake Kivu). The influence of MCS can be more obvious when the daily cycle is less pronounced. Furthermore, Zipser et al. found larger proportion of intense convection in the region corresponding to Area_sec. We rewrite the end of the discussion by referring to the work made by Zipser et al. (2006), Jackson et al. (2009). We think the differences between both areas can be explained by considering Area_max combines two conditions favorable to thunderstorm development, the convergence associated to the AEJ-S (Jackson et al., 2009) and the local effect of orography and lake.

(6) Observations with little if any interpretation

The paper has many analyses and observations that are not accompanied by physical interpretation. The Abstract for example contains no physical interpretations at all.

Table 3, Figures 3, 4 and 5 are in a similar category. This aspect needs major improvement. It is helpful if every proposed analysis has a specific scientific purpose, and so also warrants an interpretation.

Response of the authors:

An effort of development or addition of interpretation is made in abstract, discussion and conclusion. See the last sentence of the abstract.

Detailed comments/edits on the text:

The authors are not native English speakers and so there are many edits needed to clean up the text:

Line 18 Suggest dropping "very" Done

Line 20 "days" Done

Line 27 Suggest adding Williams and Satori (2004) Done

Line 28 suggest changing "space" to "spatial" Done

Line 30 change "instance" to "example"; delete "the"; "from the Lightning. . ." Done

Line 31 "resolution" Done

Line 32 "larger dynamic"? Meaning? Modified

Line 34 "maxima" Done

Line 35 change "both" to "neither" Done

Line 36 change to "maxima remains throughout the year in considering the lightning activity with 3-month seasons" Done

Line 37 what is physical interpretation of "very sharp and localized maximum" Done

Line 38 "in the eastern Democratic. . ." Done

Line 42 "scattered over a large area" Done

Line 43 "maximum activity could. . ." Done

Line 43 "linear scale for flash density was. . ." Done

Line 46 "maximum activity" Done

Line 48 change "whole" to "entire" Done

Line 49 "most of them quantified" Done

Line 51 "maximum in flash density" Done

Line 53 "The geographical extent of this region" Done

Line 57 "high spatial resolution"; "allowed a better localization and specification of its shape" Done

Line 62 "contrasting from year to year" Done

Line 63 "extends roughly" Done

Line 66 "maximum activity" Done

Line 70 change "dimension" to "area" Done

Line 77 Attributable to what? (see earlier discussion). Done

The paragraph is modified and developed.

Line 78 "the last two years" Done

Line 81 "radiation" Done

Line 84 delete the first "the" Done

Line 85 quantify "very little attenuation"; it is not small and for this reason large numbers of sensors are needed for global surveillance Done

Line 87 Why report this for 2014 when it is 2012 and 2013 that are used for analysis? Done

line 91 This would be 5 km resolution. You should justify that in terms of the accuracy of the stroke location in Africa. Done

line 94 "with the same" Done

Page 4 Done

Line 95 "the flash count" Done

Line 96 "the maximum flash density for both areas and for each year" Done

Line 97 "exhibit total flash counts" Done

Line 97-98 "indicates a stable situation from one year to the next. In contrast, the ratio..." Done

Line 99 "one year to the next" 4 digits here is overkill on precision Modified

Line 101 "localized"; "one year to the next"; "Furthermore, the spatial density..." Done

Line 103 "depends on the spatial resolution" Done

Line 104 "at a resolution" Done

Line 105 "maximum of flash density" Done

Line 109 "clearly appears" Done

Line 114 "thunderstorms, which means that the number of flashes per day is larger..." Done

Lines 115-116 These two factors could be distinguished with WWLLN observations but

you need to check the temporal development. Done

Line 117 I hope the authors disclose "specific and local conditions" Done Line 119 Which side and why? Done

Line 120 "increases markedly" Done

Line 124 "daily cycle of flashes detected by the WWLLN" Done

Line 125 "These flash counts are calculated..." Done

Line 126 "so that the flashes are associated with the ..." Done

Line 129 "for the minima in the morning..." Done

Line 130 "and for the maxima in the afternoon..." Done

Line 131 "contrast in flash counts between..." Done

Line 134 Add comma after "day" Done

Lines 133 to 136 What is your interpretation? Done

Line 138 "distribution of flashes" Done

Line 139 "year of reference"? Only one year? Done

Line 142 How were the various classes selected? Done

Line 145 "also plotted in Figure 3" (reduce redundancy) Done

Line 148 "number of days"; "about twice that of Area..."; "157 versus 84" Done

Line 155 "Variability of flash counts during..." Done

Line 157 "a clear minimum activity" Done

Line 160 "defined as the high activity" But you haven't quantified HAP and LAP. HAP and LAP are not considered anymore

Line 165 change to "and also in roughly the same proportion..." Deleted paragraph

Line 166 "with number of flashes exceeding 5000 (CL6-CL111)" Deleted paragraph

Line 167 "during the LAP" Deleted paragraph

Line 169 "during the HAP and the LAP" Deleted paragraph

Line 170 "During the HAP" Deleted paragraph

Line 171 "of days" Deleted paragraph

Line 172 "number less than 5000"; "whereas during the LAP" Deleted paragraph

Line 174 You don't have a real motivation here. Tell why you might expect correlated behavior. Done

Line 178 For this you should be giving local times, not UT times. Otherwise you lose the physical interpretation. The local time and UT time correspondence is given at section 3.2. The difference is 2 hours for this region of Africa. The local time is not always relevant. For example in Europe we have the same local time en western France and eastern Poland... The sun time is completely different within these two regions of Europe (27° of difference of longitude that is to say 1.8 hour!)

Line 180 You should be reporting correlation coefficients in the text in the same form as in the figures. Otherwise this is potentially confusing. Done

Line 182 "it also increases for the other"; "first glance" of what? The expression "at first glance" is used to express both distributions are similar after looking rapidly.

Line 190 "is more wide Done

Line 192 I don't understand the meaning here? Clear-cut? Changed

Line 193 Shouldn't this section be merged with 3.3 Annual variability. It is the same topic. The title is modified and the section is extended. Section 3.3 is different, it is an analysis day by day. No discussion of the important semiannual variation in this section. Now it is made with comments on the new graph (Figure 6c) for 2-year evolution.

Line 194 "proportion" done

Line 206 suggest adding text: "based on satellite optical observations of lightning" to distinguish from the approach taken here with VLF data. You should also define "hotspot" It is expressed like that in Albrecht et al. and not defined, so it is supposed to be understood. Maybe we can add a comment about their technique, to eliminate a 100-km in radius area around a hotspot already reported. Thus, two hotspots have at least 100 km of distance between them.

Lines 210-211 What did A. Laing say in there about MCSs?

Line 214 Considered by whom? These are not the times considered in Section 3.2. Made. It is modified for the times because we have to consider

Line 216 This is yet another time interval. More commented now.

Line 217 This is not what you reported in lines 128-130. More details are given

Line 219 "locations than our two areas"

Line 224 "result for 2011" on WWLLN ? Please clarify. Now 2011 is included to show two complete annual cycles.

Line 225 What is the meaning of "minimum proportion"? Lowest value of the proportion. It is clarified.

Line 228 The authors need to articulate their views on the ITCZ in the lightning context. In my experience, the activity lightning is usually adjacent to the ITCZ because one needs subsidence to eliminate the widespread cloudiness that is shuts off the destabilizing influence of sunlight.

236-240 Nothing is included in here about antecedent conditions of rainfall, that can influence the Bowen ratio. See also Williams and Stanfill (2002; Comptes Rendues).

Line 249 Need more discussion on the role of "great lakes" in the lightning context

Line 250 "for the development"

Line 251 "at the planetary scale"; when do "the most intense storms" max out in the diurnal cycle? Are they isolated, or are they parts of MCSs?

Line 256 "spread from the east to the western Congo basin" Done

Line 257 Only if MCS status. But don't forget role of cold outflow toward the west.

Line 259 And antecedent rainfall. In any case, more should be said about the nature of the surface in the areas selected. In this context, Williams and Satori (2004) should be consulted. Done

Line 263 "regions of strong coupling between the atmosphere..." Done

Lines 264-265 One does not want a contrast if one is seeking to explain correlated behavior. Done

Line 266 "mesoscale convective systems" Done

Line 269 "in the Congo basin" Done

Line 270 "frequently overshoots the tropopause. The climatology..." Done

Line 272 "From a five-year series of data..." Done

Line 273 "to the western side of the high mountains" Done

Line 275 "maxima in the number" Done

Line 279 I tend to agree with this statement but the discussion on MCSs needs to be elaborated on here.

Conclusions, like Abstract, is lacking in physical interpretation.

Line 282 "The spatial and temporal characteristics of the lightning..." Done

Line 282 "strongest thunderstorm activity" Done

Line 283-284 change to "with a secondary maximum"; "concentrated in the same part" Done

Line 287 to 288 "is similar in both areas Done

References

Suggest adding Williams and Satori (2004)

Williams et al. (2000, JAM) considers variations of tropical flash rates and diurnal cycles of flash rates and storm counts.

Williams (2012, Franklin Lecture) considers impact of changes in WWLLN detection efficiency over time.

Table 1 Two significant figures is probably more appropriate. In some places the authors use four! A figure is given to show the flash number in each area over the 9-year period 2005-2013. The DE values are also provided.

Figure 1 The hotspot areas straddle the equator. Some discussion is needed about that aspect alone in driving the lightning counts up high. These zones are visited at least twice per year by the zone of instability. Caption could also mention location accuracy of individual strokes.

Figure 2 Suggest changing "amounts" to "counts" Please compare this variation with those documented in Williams et al. (JAM, 2000). 4 pm is very consistent, and with

Schumann resonance observations of "background" Done.

Figure 4 Better to show flash counts that CLi classes, which require going elsewhere to check on definition/motivation. What is the thermodynamic situation on days with > CL10? Curious minds want to know.

The class is an interval of flash number. The purpose is to compare both areas and with this choice of class width, the difference is shown. We could consider more classes, but the number of flashes is displayed in Figure 5 anyway.

Figure 5 If Rˆ2 value are used here, same values should be discussed in the text. Done

Figure 6 Need more discussion on semiannual and annual variations in general. (See earlier remarks.) Done

End review Earle Williams June 23, 2017

Please also note the supplement to this comment:
https://www.nat-hazards-earth-syst-sci-discuss.net/nhess-2017-105/nhess-2017-105-AC2-supplement.pdf